# EcoDes-DK15: High-resolution ecological descriptors of vegetation and terrain derived from Denmark's national airborne laser scanning data set

Jakob J. Assmann[1], Jesper E. Moeslund[2], Urs A. Treier[1,3], Signe Normand[1,3]

[1]Department of Biology - Ecoinformatics and Biodiversity, Aarhus University, Aarhus, 8000, Denmark
[2]Department of Ecoscience - Biodiversity, Aarhus University, Rønde, 8410, Denmark
[3]Department of Biology - Center for Sustainable Landscapes Under Global Change, Aarhus University, Aarhus, 8000, Denmark

*Correspondence to*: Jakob J. Assmann (j.assmann@bio.au.dk)

**Abstract**

Biodiversity studies could strongly benefit from three-dimensional data on ecosystem structure derived from contemporary remote sensing technologies, such as Light Detection and Ranging (LiDAR). Despite the increasing availability of such data at regional and national scales, the average ecologist has been limited in accessing them due to high requirements on computing power and remote sensing knowledge. We processed Denmark's publicly available national Airborne Laser Scanning (ALS) data set acquired in 2014/15 together with the accompanying elevation model to compute 70 rasterized descriptors of interest for ecological studies. With a grain size of 10 m, these data products provide a snapshot of high-resolution measures including vegetation height, structure and density, as well as topographic descriptors including elevation, aspect, slope and wetness across more than forty thousand square kilometres covering almost all of Denmark's terrestrial surface. The resulting data set is comparatively small (~94 GB, compressed 16.8 GB) and the raster data can be readily integrated into analytical workflows in software familiar to many ecologists (GIS software, R, Python). Source code and documentation for the processing workflow are openly available via a code repository, allowing for transfer to other ALS data sets, as well as modification or re-calculation of future instances of Denmark's national ALS data set. We hope that our high-resolution ecological vegetation and terrain descriptors (EcoDes-DK15) will serve as an inspiration for the publication of further such data sets covering other countries and regions and that our rasterized data set will provide a baseline of the ecosystem structure for current and future studies of biodiversity, within Denmark and beyond. The full data set is available on Zenodo: https://doi.org/10.5281/zenodo.4756556 and a 5 MB teaser subset can be found on the GitHub code repository: https://github.com/jakobjassmann/ecodes-dk-lidar/blob/master/manuscript/figure_7/EcoDes-DK15_teaser.zip.

## 1 Introduction

Over the last decades, airborne laser scanning (ALS) has become an established data source for providing fine-resolution measures of terrain and vegetation structure in ecological research (Moeslund et al., 2019; Guo et al., 2017; Zellweger et al., 2016). Despite its informative potential and the increasing number of openly available ALS data sets with regional and national extents (Vo et al., 2016), the uptake of these data sets for large-scale ecological research and applications (such as monitoring and conservation) has remained comparatively low (Bakx et al., 2019). The low uptake is likely a consequence of the considerable challenges that remain in handling these very large data sets, which require specialist expertise and software, as well as substantial amounts of data storage and processing power (Meijer et al., 2020; Vo et al., 2016; Pfeifer et al., 2014). Here, we address this issue for Denmark by providing a compact set of ecologically relevant measures of terrain characteristics and vegetation structure derived as raster outputs from the country's national ALS data set with a grain size of 10 m x 10 m.

The typical output from an ALS survey is a so-called point cloud that describes the physical structure of the surveyed area in three-dimensional space (Bakx et al., 2019; Vierling et al., 2008). In brief, short laser pulses are sent out from a Light Detection and Ranging (LiDAR) sensor mounted on an airplane (or drone) and reflected by surfaces such as bare ground, plants or buildings. The return timing of the reflected signal is measured and - with the help of information on the sensor's orientation and position - the precise location of the reflecting surface is determined in geographic space (Vierling et al., 2008). If an object intercepting the light pulse is smaller than the beam's footprint (e.g., a leaf or a branch of a tree), some of the light may travel on and trigger a reflection from a second surface (e.g., understory vegetation or the forest floor). A single light pulse might therefore generate two or even more returns, allowing - to some degree - for the penetration of forest canopies (Ackermann, 1999). Often, the raw signal is processed by the survey provider and the resulting data is delivered to the end user in the form of a point cloud of discrete returns, where each point is associated with information on geographic location, return strength (amplitude), return number, acquisition timing etc. (Vo et al., 2016). For ALS data sets with large extents - such as Denmark's nationwide data set "DHM/Punktsky" - outputs from many survey flights are co-registered and merged, resulting in very large point clouds with hundreds of billions of points and data volumes of multiple Terabytes (Geodatastyrelsen, 2015). For further information on ALS data acquisition, we recommend Vo et al. (2016), Vierling et al. (2008) and Wagner et al. (2006).

Based on point position and neighbourhood context it is possible to separate ground and vegetation returns in ALS point clouds, allowing for the calculation of descriptors of terrain and vegetation structure. Filtering bare ground from the point cloud may be achieved with algorithms (Moudrý et al., 2020; Sithole and Vosselman, 2004), while more complex segmentation of the point clouds into object classes (such as vegetation, buildings, etc.) is done manually or with the help of supervised machine learning (see Lin et al., 2020 for a recent overview). Early applications for ALS were focussed on generating simple digital elevation models (DEMs), city and landscape planning, as well as forestry (Ackermann, 1999), but over the last decades applications have expanded into other fields, including amongst others the calculation of terrain and vegetation measures for

ecological research. Terrain derived measures of ecological interest include topographic slope, aspect (i.e., slope direction), solar irradiation, wetness etc. (e.g., Moeslund et al., 2019; Zellweger et al., 2016; Ceballos et al., 2015), and vegetation structural descriptors include vegetation density, canopy height diversity, canopy roughness and many more (e.g., Bakx et al., 2019; Moeslund et al., 2019; Coops et al., 2016). It is important to note that point cloud characteristics may limit the type of measures that can be meaningfully derived from ALS data (Bakx et al., 2019). This applies especially to the point cloud density, which needs to be high enough to meaningfully resolve the structure of understory layers in forests (Bakx et al., 2019) or ecosystems with vegetation of low stature such as grasslands or tundra (Boelman et al., 2016). Nonetheless, even simpler ALS derived descriptors of terrain and vegetation structure can be of high value for ecological applications, as fieldwork-derived alternatives are often too costly and difficult to collect over large extents (Vierling et al., 2008).

ALS data has provided critical information for research on biodiversity and habitat characteristics over the recent years, and its importance in ecological research is likely to increase in the future. Numerous biodiversity studies have successfully deployed ALS to study organisms like plants (Mao et al., 2018; Lopatin et al., 2016; Zellweger et al., 2016; Ceballos et al., 2015; Moeslund et al., 2013; Leutner et al., 2012), fungi (Peura et al., 2016; Thers et al., 2017), bryophytes, lichens (Moeslund et al., 2019), mammals (Tweedy et al., 2019; Froidevaux et al., 2016) and birds (see Bakx et al. (2019) for a comprehensive review) both in open landscapes and in forests. These studies have all emphasized the value of ALS for representing fine-scale (~ 10 m resolution) terrain or vegetation structural variation important to local biodiversity patterns. Furthermore, Valbuena et al. (2020) recently considered ALS data to be one of the key resources for deriving ecosystem morphological traits in the global assessment of Essential Biodiversity Variables (EBVs). Finding ways of making regional and nationwide ALS data more accessible to the average ecologist is therefore not only a critical priority for accelerating research on regional biodiversity patterns and species - habitat relationships, but also for the facilitation of global assessments such as those carried out by IPBES (2019) and alike.

To open up opportunities for researchers and practitioners not familiar with ALS processing or without access to the required facilities, we present a new national ALS based data set for Denmark primarily aimed at ecological research with possible uses in other disciplines. With a grain size of 10 m, these ecological descriptor (EcoDes) rasters provide a snapshot of high-resolution measures of vegetation height, structure and density, as well as topographic descriptors including elevation, aspect, slope and wetness for almost all of Denmark's terrestrial surface between spring 2014 and summer 2015 (DK15). In this publication, we a) describe the source data and outline the processing workflow (Sect. 2.1-2.3), b) summarise the data set's main characteristics (Sect. 3.1-3.2), c) describe each descriptor in detail and highlight its use and limitations (Sect. 3.3-3.4), d) provide guidance on data access and illustrate how the data could be used in an example of ecological landscape classification (Sect. 4). We finish by e) briefly discussing the general limitations of the data set and processing workflow, as well as providing perspectives on how the presented data can be complemented with other data sources (Sect. 5). We hope that ease of access

and thorough documentation of the EcoDes-DK15 data set will encourage uptake and facilitate the development of future
versions of similar data sets in Denmark and beyond.

## 2 Source data and processing workflow overview

### 2.1 Denmark - geography and ecology

Located in Northern Europe, Denmark (without Greenland and the Faroe Islands) has an approximate land area of 43 thousand
square kilometres, comprising the large peninsula of Jutland and 443 named islands. The relatively flat (highest point is 171
m above sea level) landscape predominantly consists of arable land and production forest with relatively small patches of
natural or semi-natural areas such as heathlands, grasslands, fresh and salt meadows, bogs, dunes, lakes, streams and deciduous
forests.

### 2.2 ALS and elevation source data

The Danish elevation model (DHM) is an openly available nationwide data set providing various products based on ALS data.
Here, we used the DHM/Point-cloud (DHM/Punktsky), the classified georeferenced ALS point cloud product, and the
DHM/Terrain (DHM/Terræn), the digital elevation model product derived from the point cloud. The DHM data set is currently
maintained by the Agency for Data Supply and Efficiency, Denmark (https://sdfe.dk/) and, at the time of writing, can be
downloaded from https://kortforsyningen.dk/ (continuously updated with new survey data) and https://datafordeler.dk/
(versioned). While almost all of Denmark's terrestrial surface was covered by ALS surveys in 2014/15, currently none of the
products provided by the agency contains data exclusively from these surveys. We therefore merged three different versions
of the source data to obtain a dataset that reflects the state of the vegetation in 2014/15 as best as possible, by only containing
vegetation data from 2014/15 and limited amounts from 2013 (Table 1, Sect. 3.6.3; see GitHub code repository for a detailed
description of the merger and more information on the source data sets). The DHM/Point-cloud product is a collection of 1 x
1 km tiles of three-dimensional point clouds with attributes such as position, intensity, point source ID, or classification. Point
classification follows the ASPRS LAS 1.3 standard (ASPRS, 2011), including for example ground, vegetation, and buildings.
The point density is on average 4-5 points per square meter with a horizontal and vertical accuracy of 0.15 and 0.05 metres,
respectively. Additional information on the data sets can be found in Geodatastyrelsen (Geodatastyrelsen, 2015 - in Danish),
Thers et al. (2017), Nord-Larsen et al (2017) and in the quality assessment report by Flatman et al. (2016). The DHM/Point-
cloud product is provided in LAZ-format and in the compound coordinate system for Denmark (ETRS89 / UTM zone 32N +
DVR90 height - EPSG:7416). The DHM/Terrain product is a rasterized digital model of the terrain height above sea level in
0.4 m resolution. This product is provided in a 32-bit GeoTiff format, using the same 1 km x 1 km titling convention and
spatial reference system as the DHM/Point-cloud.

**Table 1:** Overview of the data sources used for generating the EcoDes-DK15 data set. Three versions of the DHM/Pointcloud were merged to obtain a point cloud data set that contained no vegetation points scanned after 2015 and as little vegetation points before 2014 as possible. DHM/Terrain tiles were matched sources from the same data source as the corresponding point cloud tiles. A copy of the source data is archived on the internal long-term data storage at Aarhus University and is available on request. For further information see documentation on GitHub code repository and Sect. 3.6.3.

| Data source | Years | Used for | Data provider | Downloaded available from (download date) | Number of tiles |
|---|---|---|---|---|---|
| DHM/Pointcloud (DHM/Punktsky) | 2007-2018 | Vegetation Descriptors | Danish Agency for Data Supply and Efficiency | https://kortforsyningen.dk/ (24 April 2020) | 38671 |
| DHM/Pointcloud (DHM2015_punktsky) | 2007-2018 | Vegetation Descriptors | Danish Agency for Data Supply and Efficiency | https://datafordeler.dk (13 September 2020) | 10955 |
| DHM/Pointcloud (GST_2014) | 2007-2015 | Vegetation Descriptors | Danish Agency for Data Supply and Efficiency | https://kortforsyningen.dk/ (unknown, before 2017) | 47 |
| DHM/Terrain (DHM/Terræn) | 2007-2018 | Terrain Descriptors | Danish Agency for Data Supply and Efficiency | https://kortforsyningen.dk/ (24 April 2020) | 38671 |
| DHM/Terrain (DHM2015_terraen) | 2007-2018 | Terrain Descriptors | Danish Agency for Data Supply and Efficiency | https://datafordeler.dk (13 September 2020) | 10955 |
| DHM/Terrain (GST_2014) | 2007-2015 | Terrain Descriptors | Danish Agency for Data Supply and Efficiency | https://kortforsyningen.dk/ (unknown, before 2017) | 47 |

The 1 km x 1 km tiling of the DHM/Terrain 2014/2015 and DHM/Point-cloud data sets 2014/2015 match in extent and geolocation. However, a small number of tiles (n = 30) in the DHM/Point-cloud data sets did not have corresponding tiles in the DHM/Terrain data sets, these were removed prior processing resulting in the total of 49673 tiles shown in Table 1.

**2.3 Processing**

We processed the source data using OPALS 2.3.2.0 (Pfeifer et al., 2014), Python 2.7 (Van Rossum and Drake Jr, 1995), pandas 0.24.2 (Reback et al., 2019), SAGA GIS 2.3.2 (Conrad et al., 2015) from OSGgeo4W64 and GDAL 2.2.4 (GDAL/OGR contributors, 2018) also from OSgeo4W64. Some re-processing was required during the peer review process, for which we used GDAL 3.3.3 from Osgeo4W64 (GDAL/OGR contributors, 2021). The large number of tiles and descriptors to be calculated, required us to develop a robust processing pipeline, which we realised as a set of Python modules. The source code is openly available via a GitHub code repository (see Sect. 6). Processing was carried out on a Dell PowerEdge R740xd

computational server (Windows 2012 R2 64-bit Operating System, 2x Intel Xeon Platinum 8180 Processors and 1.536TB
RAM). The processing of the whole data set took approximately 45 days to complete.

2.3.1 Processing workflow
To facilitate the processing of the large data set, we first generated a set of compact Python modules providing a programming
interface that allows for the calculation of the individual descriptors outlined in Sect. 3. The individual routines were then
integrated into a Python script mediating the processing workflow in parallel, while carrying out error handling, logging and
progress tracking. The schematic of the processing workflow and the Python modules is outlined in Fig. 1. Detailed information
is available on the GitHub repository, including instructions on how to set up the processing, documentation on the functions
provided by the Python modules, as well as detailed intext commentary of the code.

We generated the processing workflow so that it should be possible to adapt it to other point cloud data sets. However, the
effort required in achieving this will vary depending on various features of the point cloud data set in question (such as tiling
and tile naming conventions, input/output grain sizes etc.). A key pre-requisite is that the point cloud is pre-classified, ideally
following the ASPRS LAS 1.1-1.4 standards (ASPRS, 2019). We have also provided a helper script that can be adapted to
generate a raster DTM from the point cloud should this not be available, see the documentation on the GitHub repository for
the details. Finally, the modular nature of the processing workflow allows for only a subset of the output descriptors to be
calculated and the integration of additional processing routines for any new user-defined descriptors.

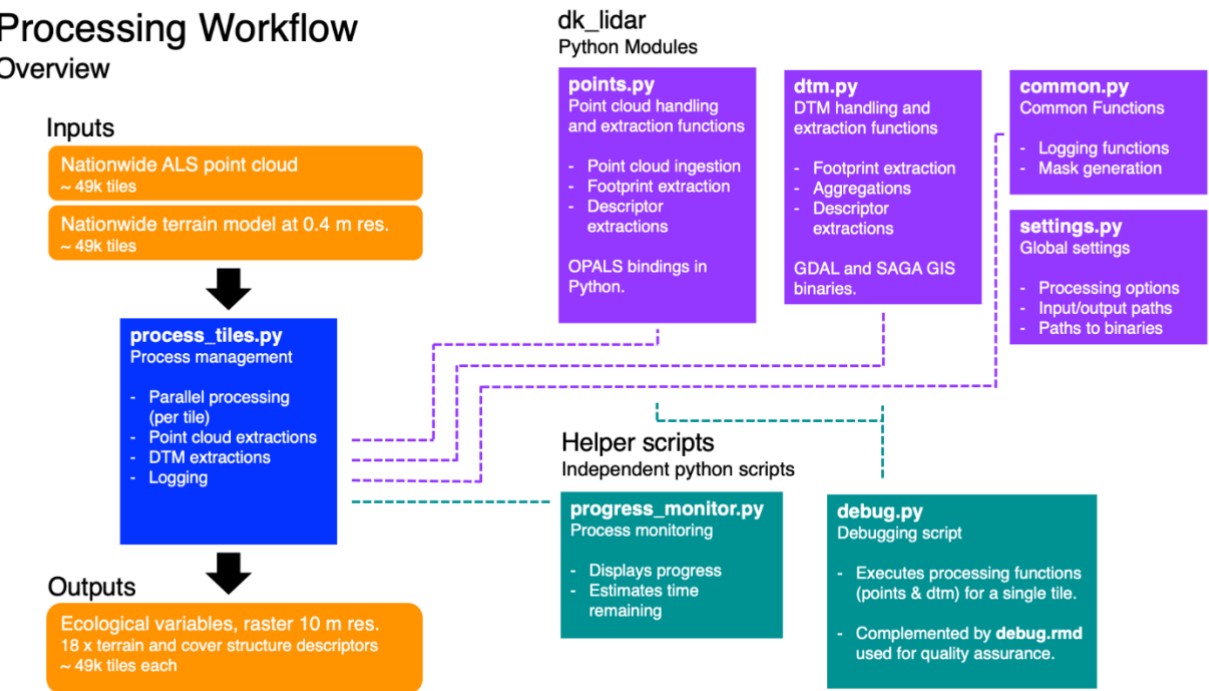


**Figure 1:** Diagram of the processing workflow, the *dk_lidar* Python module and helper scripts. The workflow requires two inputs: a pre-classified set of point cloud tiles and a paired set of digital terrain model (dtm) tiles. The process management is handled by the *process_tiles.py* script which facilitates processing of each tile pair (dtm and point cloud) in parallel and logs the progress. For each tile, *process_tiles.py* calls a specified set of extraction and processing functions from the *dk_lidar* modules. Point cloud extraction functions are specified in *points.py* and terrain model extraction functions are specified in *dtm.py*. The *dk_lidar* modules also contain two further files, *common.py* a script containing specifications of common functions used by the *points.py* and *dtm.py,* as well as *settings.py* which is used to set global processing options, specify file paths etc. Finally, two helper scripts are provided *progress_monitor.py* which facilitates progress monitoring and estimates the time remaining and *debug.py* a script for testing the workflow for a single tile. Together the Python scripts and modules allow to generate the ecological descriptor outputs from the two input data sets. Further documentation of the dk_lidar modules and workflow scripts can be found on the GitHub repository associated with this publication: https://github.com/jakobjassmann/ecodes-dk-lidar.

## 3 Data set description and known limitations

### 3.1 Extent, projection, resolution and data format

EcoDes-DK15 covers the majority of Denmark's land area, including the island of Bornholm (approximate extent: 54.56 °N to 57.75 °N, 8.07 °E to 15.20 °E). The data is projected in ETRS89 UTM 32 N based on the GRS80 spheroid (EPSG: 25832). The data set is available as GeoTIFFs with 10 m grain size via a data repository on Zenodo (see Sect. 6). For each descriptor the nation-wide data are split into 49673 raster tiles of 1 km x 1 km with a 10 m grain size based on 25-fold aggregations of the 0.4 m national grid of Denmark. A virtual raster mosaic (VRT) file is provided for each descriptor (except the point_source_counts, point_source_ids and point_source_proportion descriptors), and a file containing the tile footprint geometries can be used for geographical sub-setting of the data. We also provide masks for inland water and the sea.

The final data set consists of just under 94 GB of data (compressed for download 16.8 GB). To reduce the size of the data set we converted numerical values from floating point precision to 16-bit integers where possible. In some cases, this required us to stretch the values by a set factor to maintain information content beyond the decimal point. The descriptor conversion factors are available as a csv file provided with the data set and in Table 2. Missing data (NoData) is denoted by a value of -9999 throughout the data set.

### 3.2 Overview and file naming convention

An overview of the eighteen terrain and vegetation structure descriptors as well as the auxiliary data provided can be found in Table 2. Generally, the descriptor names in Table 2 reflect the prefix of the file name of a GeoTiff file within the data set. This prefix is followed by a suffix representing the unique identifier for each tile based on the UTM coordinates of the tile (see Sect. 3.4.3 for more detail). When working with the complete data set, tiles from the same descriptor are grouped within a folder using the same descriptor name as used for the file name prefix. For example, for the tile with the unique id "6239_446" the GeoTiff for the "dtm_10m" descriptor can be found in "dtm_10m/dtm_10m_6239_446.tif". The exceptions are the point counts, vegetation proportions and point source information, please see the relevant sections below for more detail.

**Table 2:** Brief overview of the eighteen main EcoDes-DK15 descriptors and descriptor groups, their ecological meaning, unit, format and conversion factor. See Sect. 3.4 for a detailed description of each descriptor. In addition to the 70 raster layers for the main descriptors, the data set contains nine layers of auxiliary information (see Sect. 3.7). Note: to obtain the correct unit, the descriptor value needs to be divided by the conversion factor.

| Descriptor(s) | Ecological meaning | Unit | Format | Conversion factor | Number of layers |
|---|---|---|---|---|---|
| dtm_10m | elevation | m | 16-bit integer | 100 | 1 |

| aspect | topographic aspect | degrees | 16-bit integer | 10 | 1 |
|---|---|---|---|---|---|
| slope | topographic slope | degrees | 16-bit integer | 10 | 1 |
| heat_load_index | proxy of radiation and wetness | unitless | 16-bit integer | 10000 | 1 |
| solar_radiation | solar radiation | MJ x $100^{-1}$ $m^{-2}$ x $yr^{-1}$ | 32-bit integer | 1 | 1 |
| openness_mean | topographic position | degrees | 16-bit integer | 1 | 1 |
| openness_difference | presence of linear landscape features | degrees | 16-bit integer | 1 | 1 |
| twi | topographic wetness | unitless | 16-bit integer | 1000 | 1 |
| | | | | | |
| amplitude_mean | complex** | undefined | 32-bit float | 1 | 1 |
| amplitude_sd | complex** | undefined | 32-bit float | 1 | 1 |
| canopy_height | vegetation height | m | 16-bit integer | 100 | 1 |
| normalized_z_mean | average structural height (incl. vegetation and buildings) | m | 16-bit integer | 100 | 1 |
| normalized_z_sd | variation in structural height (incl. vegetation and buildings) | m | 16-bit integer | 100 | 1 |
| point_counts* | number of returns in ground, water, building and vegetation point classes; total return count and vegetation return counts in height bins | count | 16-bit integer | 1 | 30 |
| vegetation_proportion* | proportion of vegetation returns in height bins | proportion | 16-bit integer | 10000 | 24 |
| vegetation_density | ratio of vegetation returns to total returns | proportion | 16-bit integer | 10000 | 1 |
| canopy_openness | ratio of ground and water returns to total returns | proportion | 16-bit integer | 10000 | 1 |
| building_proportion | ratio of building returns to total returns | | 16-bit integer | 10000 | 1 |
| | | | | | |
| point_source_info* | point source / flight strip information | varied, see description | varied, see description | varied, see description | 4 |
| masks | inland water and sea mask | binary | 16-bit integer | 1 | 2 |
| date_stamp* | min, max and mode of GPS dates for all vegetation points | date as YYYYMMDD*** | 32-bit integer | 1 | 3 |


* Descriptor group containing multiple individual descriptors, see intext description for detail.
** The amplitude descriptors are difficult to interpret, but can serve as useful indicators for vegetation classification and
biodiversity studies. Please see intext description for more detail.
*** YYYY = year in four digits, MM = month in two digits, DD = day in two digits.
**3.3 Completeness of the data set**
The processing of the data set was almost completely successful. Processing failed on average for only 18 out of the 49673
tiles per descriptor with a maximum of 65 tiles failing for the *canoy_height*, *normalized_z_mean* and *normalized_z_sd*
descriptors. The majority of these tiles were located on the fringes of the data set, including sand spits, sandbanks etc, we
therefore did not attempt re-processing of those tiles. Instead, we generated NoData rasters for all missing descriptor - tile
combinations (i.e. we assigned -9999 to all cells in those tiles). We provide a text file listing the affected "NoData" tiles in the
folder of each descriptor (the file is named empty_tiles_XXX.txt, where XXX is the descriptor name).
**3.4 Elevation-model derived descriptors**
The following descriptors were solely derived from the 0.4 m digital elevation model (DHM/Terrain). Visualisations of these
descriptors for an example tile in the Mols Bjerge area are shown in Fig. 2.

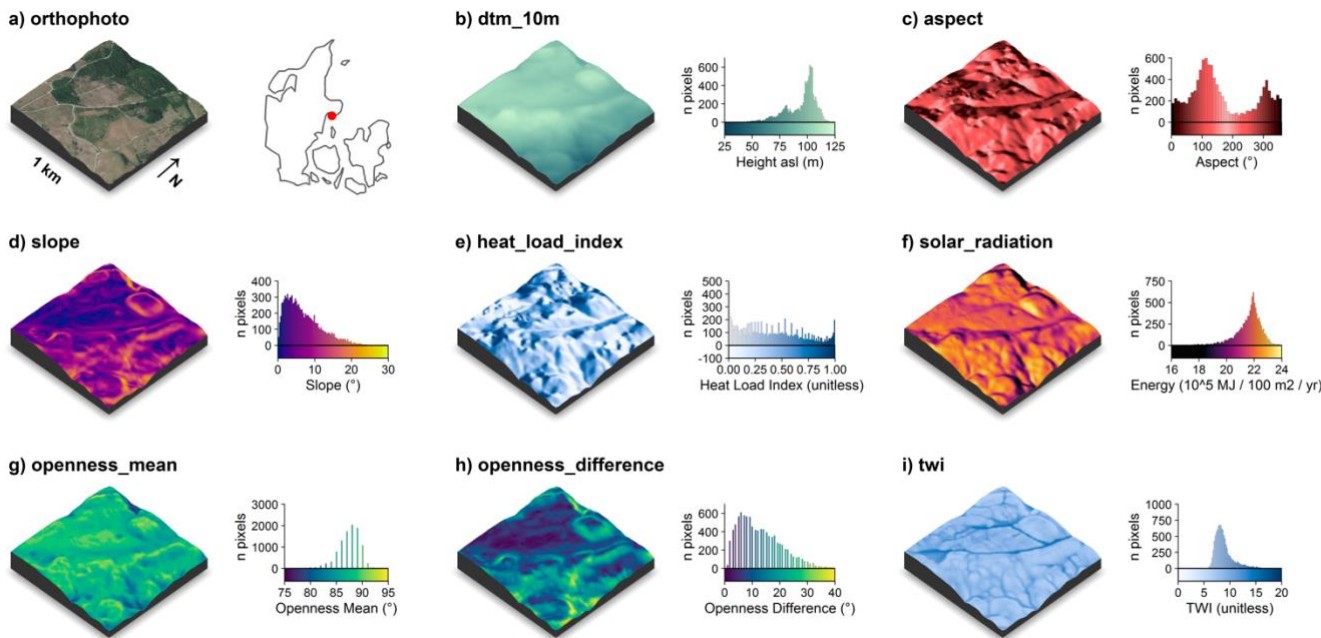


**Figure 2:** Illustration of the terrain model derived descriptors for a 1 km x 1 km tile in the Mols Bjerge area (tile id: 6230_595).
An orthophoto and the tile location relative to Denmark are shown in (a). The terrain model (dtm_10m) is illustrated in (b).
The terrain derived descriptors comprise of: c) the topographic aspect, d) the topographic slope, e) the heat load index following
Kuehne et al. f) the estimated incident solar radiation, g) the landscape openness mean, h) the landscape openness difference
in the eight cardinal directions and i) the topographic wetness index (TWI) based on Kopecký et al. (2020). For visualisation
purposes, we amplified the altitude above sea-level by a factor of two in the 3D visualisations and divided the solar radiation
values by $10^5$. The 3D raster visualisations were generated using the rayshader v0.19.2 package in R (Morgan-Wall, 2020).
Orthophoto provided by the Danish Agency for Data Supply and Efficiency (https://sdfe.dk/hent-data/fotos-og-geodanmark-
data/).

### 3.4.1 Elevation (dtm_10m)

We aggregated the 0.4 m DEM by mean to match the 10 m x 10 m national grid of the remainder of the data set. We used
*gdalwarp* to carry out the aggregations. Values represent the elevation above sea level in metres (DVR90, EPSG: 5799)
multiplied by a factor of 100, rounded to the nearest integer and converted to 16-bit integer.

### 3.4.2 Aspect (aspect)

The topographic aspect describes the orientation of a slope in the terrain and may, amongst other things, be related to plant
growth through light and moisture availability. We calculated the aspect in degrees, with 0° indicating North, 90° East, 180°
South and 270° West. Values represent the aspect derived from a 10 m aggregate of the elevation model (aggregated by mean
with 32-bit floating point precision). Calculations were carried out using *gdaldem* binaries and the "aspect" option, which by
default uses Horn's method to calculate the aspect (Horn, 1981). To avoid edge effects, all calculations were done on a mosaic
that included the focal tile and all available directly neighbouring tiles (maximum eight). The mosaic was cropped back to the
extent of the focal tile upon completion of the calculations. We then converted the value for each cell from radian to degrees,
multiplied it by a factor of 10, rounded to the nearest integer and stored the results as a 16-bit integer. Finally, we assigned a
value of -10 (-1°) to all cells where the slope was 0° (flat). Limitations in the aspect arise in relation to edge effects that occur
where a neighbourhood mosaic is incomplete for a focal tile (i.e., less than eight neighbouring tiles), such as for tiles along the
coastline or at the edge of the covered extent. For those tiles, no aspect can be derived for the rows or columns at the edge of
the mosaic. The cells in those rows and columns have no neighbouring cells and were assigned the NoData value (-9999).
Please also note that we calculated the aspect descriptor from the 10 m aggregate of the DTM/Terrain data set rather than
deriving it from the 0.4 m original resolution rasters and then aggregating it. The latter approach could represent the
aspect/slope at the original resolution better (Grohmann, 2015; Moudrý et al., 2019), but would create inconsistencies within
how the remaining DTM/Terrain descriptors are calculated in this dataset.

### 3.4.3 Slope (slope)

The topographic slope describes the steepness of the terrain and amongst other things may be related to moisture availability,
exposure and erosion. We derived the topographic slope in degrees with a 10 m grain size from a mean aggregate of the
elevation model (32-bit floating point precision) using the *gdaldem* binaries with the "slope" option, which by default use
Horn's method to calculate the slope (Horn, 1981). To avoid edge effects, we carried out the calculations on a mosaic including
the focal tile and all available directly neighbouring tiles (maximum eight). The mosaic was cropped back to the extent of the
focal tile upon completion of the calculations. The value for each cell was converted from radian to degrees, multiplied by a
factor of 10, rounded to the nearest integer and stored as a 16-bit integer. Limitations in the slope arise in relation to edge
effects that occur where a neighbourhood mosaic is incomplete for a focal tile (i.e., less than eight neighbouring tiles), such as
for tiles along the coastline or at the edge of the covered extent. For those tiles, no slope can be derived for the rows or columns
at the edge of the mosaic. These cells in those rows and columns have no neighbouring cells and *gdaldem* assigns the NoData
value (-9999) to these cells. Please also note that we calculated the slope descriptor from the 10 m aggregate of the
DTM/Terrain data set rather than deriving it from the 0.4 m original resolution rasters and then aggregating it. The latter
approach could represent the aspect/slope at the original resolution better (Grohmann, 2015; Moudrý et al., 2019), but would
create inconsistencies within how the remaining DTM/Terrain descriptors are calculated in this dataset.

### 3.4.4 Landscape openness mean (openness_mean)

Landscape openness is a landform descriptor that indicates whether a cell is located in a depression or elevation of the
landscape. We calculate the landscape openness following Yokoyama et al. (2002) using the OPALS implemented algorithms.
We used a mean aggregate of the elevation model with 10 m grain size and 32-bit floating point precision, and derived the
mean landscape openness for a cell as the mean of the landscape openness in all eight cardinal directions with a search radius
of 150 m. We chose to base this descriptor on the aggregated 10 m elevation model and a 150 m search radius, as we think
that these are best suited to describe the landscape scale variation in the landforms of Denmark. Danish landscapes are
characterised by gently undulating terrain, valleys forged by small to medium sized rivers and dune systems along the
coastlines. First, we generated a mosaic including the focal tile and all available tiles in the direct neighbourhood (max. eight
neighbouring tiles) to reduce edge effects in subsequent calculations. The mean of the positive openness for all eight cardinal
directions with search radius of 150 m was then derived for all cells in the mosaic using the OPALS Openness module (options:
feature = 'positive', kernelSize = 15 and selMode = 0). Next, the mean openness per cell was converted from radians to degrees,
rounded to the nearest integer and stored as a 16-bit integer. For incomplete neighbourhood mosaics (i.e. containing less than
eight neighbouring tiles) we then masked out cells within the first 150 m of all edges where a neighbourhood tile was missing.
Finally, the output was cropped back to the extent of the focal tile. As a consequence of the edge effect related masking, the
focal tiles on the fringes of the data set, such as those on coastlines or at the edge of the coverage area, have no data available
for the first 150 m. The corresponding cells for the affected areas are set to the NoData value -9999.

### 3.4.5 Landscape openness difference (openness_difference)

In addition to the mean of the landscape openness, we also derived a landscape openness difference measure. This difference
measure is an indicator of whether a cell is part of a linear feature in the landscape that runs in one cardinal direction, such as

a ridge or valley, therefore providing additional information to the landscape openness_mean descriptor. We calculated the landscape openness difference based on the 10 m mean aggregate of the elevation model (32-bit floating point precision) and with a search radius of 50 m. We chose these parameters as we consider them best suited to capture the relatively narrow valleys and ridgetops common in the Danish landscape. First, we generated a mosaic including the focal tile and all available tiles in the direct neighbourhood (max. eight neighbouring tiles) to reduce edge effects in subsequent calculations. We then calculated the minimum and maximum of the positive landscape openness from all eight cardinal directions for all cells in the mosaic using the OPALS Openness module with a search radius of 50 m (feature = 'positive', kernelSize = 5 , selMode = 1 for minimum and selMode = 2 for maximum). Next, we converted the minimum and maximum values from radian to degrees and calculated the difference between the maximum and minimum value. We rounded the result to the nearest full degree. For the cases where the neighbourhood mosaic was incomplete, i.e., containing less than eight neighbouring tiles, we masked out all cells within the first 50 m of all edges with a missing neighbourhood tile. The final output mosaic was then cropped to the extent of the focal tile and stored as a 16-bit integer GeoTIFF. As a consequence of the edge effect related masking, focal tiles on the edges of the data set, such as those on coastlines or at the edge of the coverage area, have no data available for the first 50 m.

### 3.4.6 Solar Radiation (solar_radiation)

Incident solar radiation is a key parameter for plant growth as it represents the electromagnetic energy available to plants required for photosynthesis. However, in the comparatively flat country of Denmark, shading by other vegetation likely exerts a larger influence on photosynthetic activity than terrain related shading. Here, the impact of incident solar radiation on the local climate likely plays a more important role for determining plant growth due to its influence on drought/water dynamics (Moeslund et al., 2019). We estimated the amount of incident solar radiation received per cell (100 $m^2$) per year from the slope and aspect computed as described above. Calculations were implemented using *gdal_calc*, following equation 3 specified in McCune and Keon (2002):

$$solar\_radiation = 10^{\wedge}6 \times e^{0.339+0.808 \times \cos(L) \times \cos(S) - 0.196 \times \sin(L) \times \sin(S) - 0.482 \times cos(180 - |(180 - A)|) \times sin(S)} \qquad (1)$$

where *L* is the centre latitude of the cell in degrees, *S* is the slope of the cell in degrees and *A* is the aspect of the cell in degrees. The resulting estimate is given in: MJ x $100^{-1}$ $m^{-2}$ x $yr^{-1}$ (McCune and Keon, 2002). Slope and aspect for each 10 m x 10 m grid cell were sourced from the slope and aspect rasters. We saved the result as 32-bit integers. Due to propagation from the calculation of slope descriptor, no solar radiation values can be calculated for cells found right on the edge of the data set, for example in tiles situated along the coastline or at the edge of the sampling extent.

**3.4.7 Heat Load Index (heat_load_index)**

The heat load index (McCune and Keon, 2002) was originally developed as an indicator for temperature based solely on aspect, but this characteristic is probably better captured in our solar radiation descriptor (see above) that was developed to improve shortcomings in the heat load index (McCune and Keon, 2002). However, in a previous study (Moeslund et al., 2019) we show that - in Denmark - the index was moderately correlated with soil moisture, and can therefore serve as a useful indicator of the amount of moisture available to plants. We calculated the heat load index based on the aspect rasters (described above) following the equation specified in McCune and Keon (2002) using *gdal_calc*:

$$heat\_load\_index = \frac{(1 - cos(A - 45))}{2}$$

(2)

where *A* is the aspect in degrees. We stretched the result by a factor of 10000, rounded to the nearest integer and stored it as a 16-bit integer. As the heat_load_index is not meaningfully defined for flat cells (slope = 0° / aspect = -1°), we set the value of those cells to NoData (-9999). Finally, for cells that are located on the outermost edges of the data set the heat_load_index is not defined due to propagation of the NoData value assigned to the aspect in those cells.

**3.4.8 Topographic wetness index (twi)**

The topographic wetness index (TWI) provides a proxy measure of soil moisture or wetness based on the hydrological flow modelled through a digital terrain model. Here, we derived the TWI following the method recommended by Kopecký et al. (2020). We based our calculations on the aggregated 10 m elevation model (dtm_10m, 16-bit integer) and used a neighbourhood mosaic (max. 8 neighbours) for each focal tile to derive the TWI. The exact procedure is detailed in the next paragraph. As such the index values calculated by us only consider a catchment the size of one tile and all its neighbours (for non-edge tiles this is a 3 km x 3 km catchment, for edge tiles it is smaller depending on the completeness of the neighbourhood mosaic). We then cropped the resulting output back to the extent of the focal tile, stretched the TWI values by a factor of 1000, rounded to the next full integer and stored the results as a 16-bit integer.

We calculated the TWI using SAGA GIS v. 7.8.2 binaries. First, we sink-filled the neighbourhood mosaic of the terrain model using the *ta_preprocessor 5* module and the option "MINSLOPE 0.01" (Wang and Liu, 2006). Second, we calculated the flow accumulation based on the sink-filled neighbourhood mosaic of the terrain model (from step one) using the *ta_hydrology 0* module with options "METHOD 4" and "CONVERGENCE 1.0" (Freeman, 1991; Quinn et al., 1991). Third, we derived the flow width and specific catchment area based on the sink-filled neighbourhood mosaic of the terrain model (from step one) and the flow accumulation (from step two) using the module *ta_hydrology 19* (Gruber and Peckahm, 2008; Quinn et al., 1991). Fourth, we calculated the slope based on the sink-filled neighbourhood mosaic of the terrain model (from step one) using the *ta_morphometry 0* module with option "METHOD 7" (Haralick, 1983). Finally, we derived the TWI based on the specific

catchment area (from step three) and slope (from step four) using the module ta_hydrology 20 (Beven and Kirkby, 1979; Böhner and Selige, 2006; Moore et al., 1991). For detailed descriptions of the modules used, please refer to the SAGA GIS documentation (SAGA-GIS Tool Library Documentation v7.8.2, 2021).

The TWI descriptor calculated for EcoDes-DK15 is subject to two main limitations: edge effects and small catchment size. Tiles with incomplete neighbourhoods (i.e., less than 8 direct neighbours are available) will suffer from edge effects in the direct vicinity of the relevant border and overall due to a reduced catchment size. Furthermore, even in the ideal case of the neighbourhood being complete, for most cells flow accumulation is therefore only calculated for the direct neighbourhood of a focal tile, comprising a 3 km x 3 km catchment area. While we hypothesize that, due to the relatively low variation in topography in Denmark, the TWI based on this comparably small catchment area will serve as a reasonable proxy for terrain-based wetness in most cases, it may be less reliable in areas with exceptionally high variation in topography or for lakes and rivers with large catchments. In addition, we would like to point the reader towards the general limitations of the TWI as a proxy for soil moisture or terrain wetness as for example discussed by Kopecký et al. (2020). These general limitations should be taken into account when interpreting the TWI values provided in EcoDes-DK15.

**3.5 Point-cloud derived descriptors**

The DHM/Point-cloud point cloud was pre-classified into eleven point categories (Geodatastyrelsen, 2015) following the ASPRS LAS 1.3 standard (ASPRS, 2011). For the EcoDes-DK15 data set, we restricted the analysis to six of these classes, including ground points ("Terræn") - class 2, water points ("Vand") - class 9,  building points ("Bygninger") - class 6, as well as low ("lav"), medium ("mellemhøj") and high vegetation ("høj vegetation") - classes 3, 4 and 5, respectively. We grouped the three vegetation classes into one single vegetation class and, instead of the pre-assigned height categories, considered a more detailed set of height bins (see point count and proportion descriptions below). The overall classification accuracy of the point cloud was assessed by the Danish authorities (Flatman et al., 2016), but limited information is available for the accuracy in each class. Thus, some degree of noise should be assumed across all classes. The tall vegetation category (class 6) was used as a catch-all class if classification failed, as often the case for very tall buildings and structures (Flatman et al., 2016). To reduce the noise related to such structures, we removed vegetation points with a normalised height exceeding 50 m above ground when calculating the vegetation point counts. We included all returns, i.e., first returns and echoes, in our analysis.

All point cloud processing was carried out using OPALS and the OPALS Python bindings. As none of the point cloud derived descriptors required mosaicking to prevent edge-effects, we processed all point cloud descriptors on the focal tile only. After the initial ingestion of the LAZ-file for a tile into the OPALS data manager format (odm), we used the *OpalsAddInfo* module to add a normalised height (z) attribute to the points. For this attribute we subtracted the height of the ground derived from the corresponding DHM/Terrain raster (0.4 m grid size) from the height above sea level of each point. Figure 3 illustrates how the

372 point cloud data translates to some of the descriptor outputs for four exemplary 10 m x 10 m cells from the data set, and an
373 overview of the point cloud derived descriptors for a 1 km x 1 km tile in Vejle Fjord in central Jutland is provided in Fig. 4.

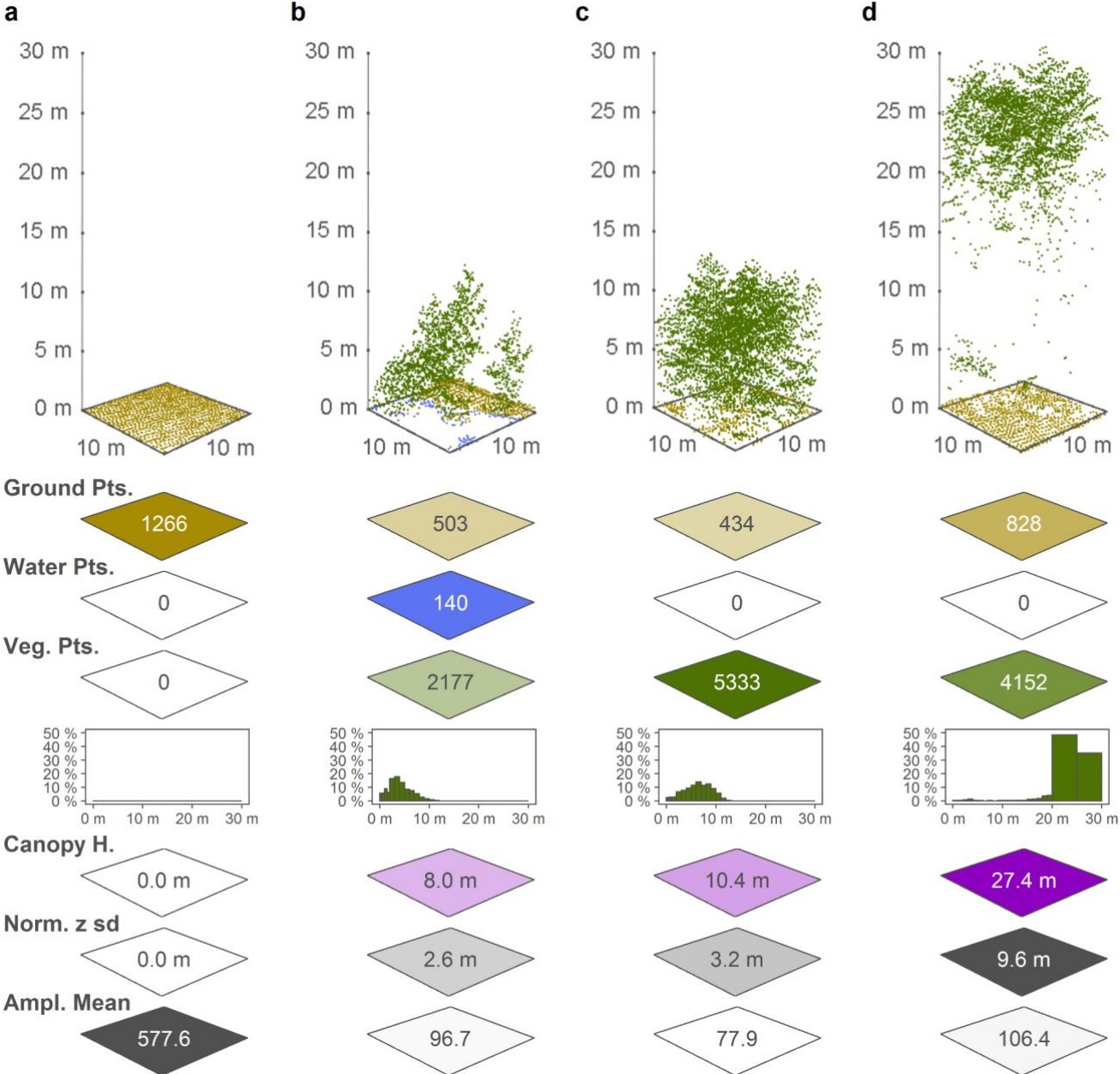

**Figure 3:** Point cloud examples for four 10 m x 10 m cells and a selection of the associated EcoDes-DK15 descriptors derived
from the point clouds, illustrating the ecological meaning and some of the limitations of the EcoDes-DK15 data set. The 10 m
377 x 10 m cells represent the following environments: a) an agricultural field, b) the edge of a forest / parkland pond with low
vegetation, c) a young plantation of dense coniferous trees, and d) old growth mixed-woodland. The EcoDes-DK15 descriptors

shown include (from the top) the total point counts for each cell in the three main EcoDes-DK15 categories: 1) the number of returns classified as ground, 2) the number of returns classified as water and 3) the number of returns classified as vegetation. In addition, the relative proportion of vegetation points per predefined height bin is illustrated below the total vegetation point count. Finally, the bottom three panels show the estimated canopy height (altitude above ground for the 95%-percentile of all vegetation returns), the normalized z standard deviation (variation in height above ground for all return classes), and the mean return amplitude for each cell. Please note how the classification of the point cloud classification does not separate between very low growing vegetation (e.g., grass) and ground points in the agricultural field shown in a), and how returns from water are only registered in shallow areas close to the water bodies edge, such as exemplified by the forest pond in b). Lastly, we would like to point the reader to the general limitations of ALS in penetrating forest canopies such as those shown in c) and d). While the upper layers of the canopies are well resolved in both cases, the laser scanning struggles to capture some aspects of the lower layers; the ground returns were frequently blocked by the thick canopy in c) and the laser fails to meaningfully characterise understory vegetation and stems in d).

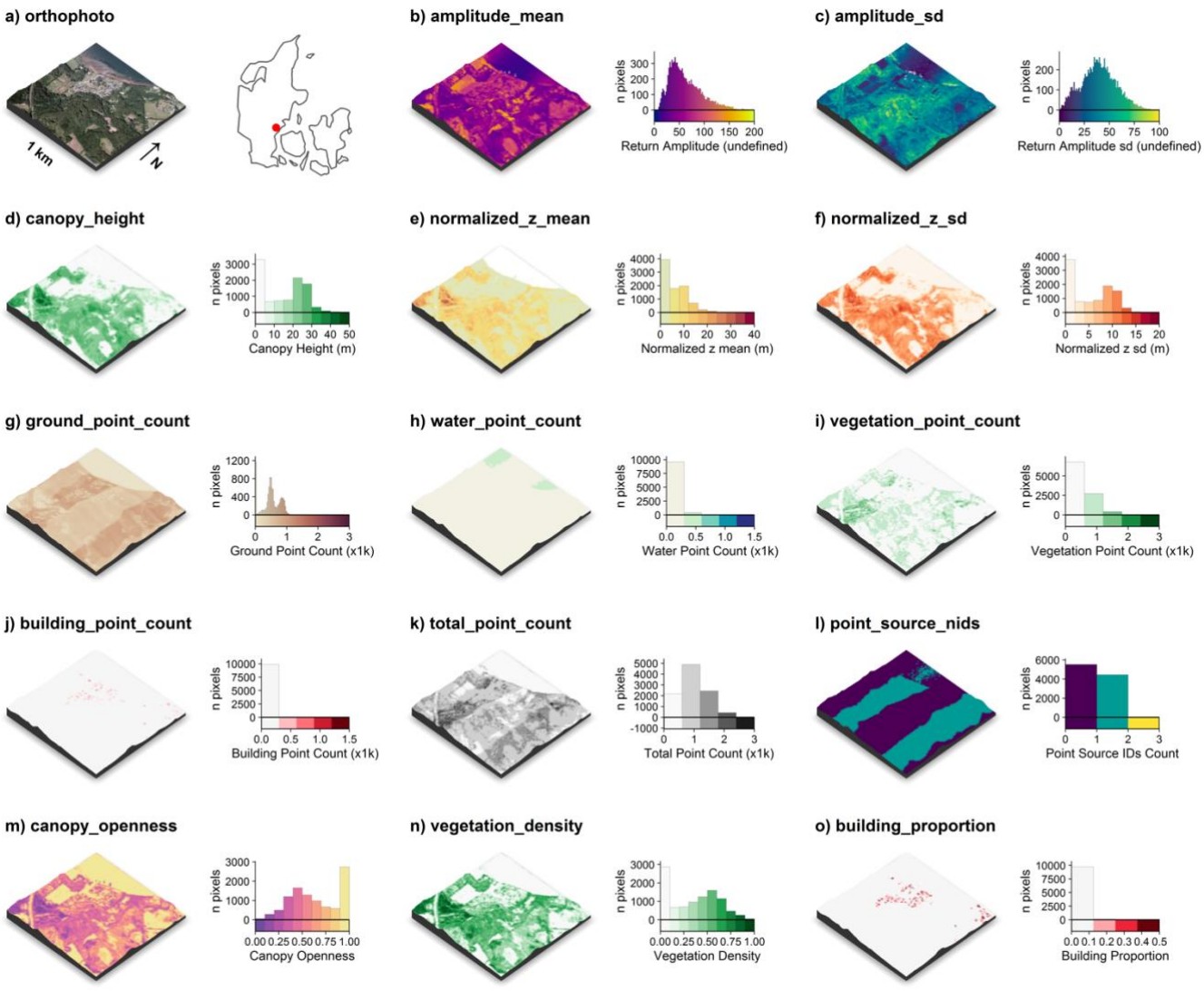

**Figure 4:** Illustration of the point cloud derived descriptors for a 1 km x 1 km tile along Vejle Fjord (tile id: 6171_541). An orthophoto and the tile location relative to Denmark are shown in (a). The point cloud derived descriptors comprise of: c) the mean return amplitude, d) the standard deviation in the return amplitude, e) the canopy height (vegetation returns only), f) the mean of the normalized height above ground (all returns), g) the mean of the normalized height (all returns), h) the ground point count, i) the water point count, j) the building point count, k) the total point count, l) the number of point sources (flight strips), m) the canopy openness, n) the vegetation density and o) the building proportion. Note the influence of point source overlap illustrated in l) on some of the descriptors, for example: g) ground point count, i) vegetation point count and k) total point count (see Sect. 3.5.5 for detail). For visualisation purposes, we amplified the altitude above sea-level by a factor of two in the 3D visualisations and divided the point counts by 1000. The 3D raster visualisations were generated using the rayshader

v0.19.2 package in R (Morgan-Wall, 2020). Orthophotograph provided by the Danish Agency for Data Supply and Efficiency
(https://sdfe.dk/hent-data/fotos-og-geodanmark-data/).

**3.5.1 Amplitude – mean and standard deviation (amplitude_mean and amplitude_sd)**

The amplitude attribute of a point in the DHM/Point-cloud is the actual amplitude of the return echoes, i.e., it describes the strength of the LiDAR return signals detected by the sensor. The descriptor is difficult to interpret in terms of its ecological meaning. Nonetheless, we believe that it is still useful for vegetation classifications, biodiversity analysis and other applications that perform well with proxy data. We calculate the arithmetic mean and standard deviation of the amplitude for all points within a 10 m x 10 m cell. Here, 'all points' refers to all points classified as ground, water, building, and vegetation points. Calculations were carried using the *OPALS Cell* module and results were stored as 32-bit floats. The amplitude attributes in the DHM/Point-cloud point clouds are not directly comparable when points originate from different point sources (e.g., flight strips), as the amplitude has not been calibrated and hence is sensitive to differences in sensor, sensor configuration and signal processing. Calculating summary metrics such as mean and standard deviation for a 10 m x 10 m cell where points from different point sources are present introduces additional complexities. In some cases, a 10 m cell might contain points from up to four different sources. We therefore recommend using the two amplitude descriptors with care, and - if possible - in conjunction with information on the point source ids contained in the point_source_info descriptors described below.

**3.5.3 Canopy height (canopy_height)**

Canopy height is a key parameter of vegetation structure related to biomass and ecosystem functioning. We derived the canopy height in metres as the 95th-percentile of the normalised height above ground of all vegetation points within each 10 m x 10 m cell using the *OPALS Cell* module. The resulting canopy heights were multiplied by a factor of 100, rounded to the nearest integer and stored as 16-bit integers. In cases where there were no vegetation points in any given cell, we set the canopy height value of the cell to zero metres. Please note that the canopy height is therefore also set as zero metres even if there are no points present in the cell at all (such as ground or water points). Furthermore, our algorithm calculates the canopy height even if there is only a small amount of vegetation points in a cell. In rare cases, this might lead to erroneous canopy-height readings if vegetation is found on artificial structures or points have been mis-classified. For example: A tall communications tower can be found just south of Aarhus and returns from the tower were miss-classified as vegetation. The resulting canopy height for this cell is calculated as > 100 m above ground, which would not make sense if interpreted as a height of the vegetation above ground. For such cases, the building proportion descriptor may be used to separate cells with artificial structure from those with vegetation only. See also the "normalized_z" descriptor below for a closely related measure.

**3.5.4 Normalised height - mean and standard deviation (normalized_z_mean and normalized_z_sd)**

Similar to the canopy height descriptor, the normalised height describes the structure properties of the point cloud above ground. The key difference between the two descriptors is that for the normalised height we also included non-vegetation

points (buildings & ground) and derived the summary statistic as the mean rather than the 95%-quantile. For the normalised height descriptor, we also provide a measure of variation in form of the standard deviation. Specifically, we calculated the normalised mean and the standard deviation of the mean height above ground (normalised z attribute) for all points in each 10 m x 10 m grid cell using the *OPALS Cell* module. The results were multiplied by 100, rounded to the nearest integer and stored as 16-bit integers. We used the normalised z attribute generated during the ingestion of the point cloud reflecting the height of a point relative to the ground level determined by the DHM/Terrain raster. Here, all points refer to all points belonging either to the ground, water, building or vegetation class. By definition the normalised height mean will be highly correlated with the "canopy_height" descriptor for cells where mainly vegetation points are present. We kept the American spelling of the descriptor name for legacy reasons with previous versions of the data set.

### 3.5.5 Point counts (xxx_point_count_xxx)

The point count descriptors are intermediate descriptors used to generate the proportion descriptors described below. However, they can also be used to calculate tailored proportion descriptors relevant to addressing a specific ecological objective (see use-case example in Sect. 4.2). For EcoDes-DK15 we derived thirty point count descriptors for each 10 m x 10 m cell based on filtering of the pre-defined point classifications and separation by height above ground (normalised z) using the *OPALS Cell* module. All point counts were stored as 16-bit integers. These thirty descriptors contain six general point counts, including ground, water, vegetation, building and total point counts (Table 3), as well as twenty-four vegetation point counts separated in height bins (Table 4). Note that the number of returns within a 10 m cell is influenced by a) the number of point sources present in the cell, b) the relative position and distance of a cell to the point source when the data was collected (i.e., to the flight path), and c) by the point source themselves (i.e., differences between the LiDAR sensors deployed). The absolute counts are therefore not directly comparable between cells and need to be standardised first, for example by division of the total number of point counts as done for the point proportion descriptors derived by us.

**Table 3:** General point count descriptors, as well as the height ranges and point classes included in each descriptor.

| Descriptor name | Height range | Point classes |
|---|---|---|
| ground_point_count_-01m-01m | -1 m to 1 m | ground points (class 2) |
| water_point_count_-01m-01m | -1 m to 1 m | water points (class 9) |
| ground_and_water_point_count_-01m-01m | -1 m to 1 m | ground and water points (classes 2,9) |
| vegetation_point_count_00m-50m | 0 m to 50 m | vegetation points (classes 3,4,5) |
| building_point_count_-01m-50m | -1 m to 50 m | building points (class 6) |
| total_point_count_-01m-50m | -1 m to 50 m | ground, water, vegetation and building points (classes 2,3,4,5,6,9) |

**Table 4:** Vegetation point count descriptors divided into twenty-four height bins. All vegetation point counts include the point
classes 3,4 and 5.

460

| Descriptor name | Height range |
| --- | --- |
| vegetation_point_count_00.0m-00.5m | 0.0 m to 0.5 m |
| vegetation_point_count_00.5m-01.0m | 0.5 m to 1.0 m |
| vegetation_point_count_01.0m-01.5m | 1.0 m to 1.5 m |
| vegetation_point_count_01.5m-02.0m | 1.5 m to 2.0 m |
| vegetation_point_count_02m-03m | 2 m to 3 m |
| vegetation_point_count_03m-04m | 3 m to 4 m |
| vegetation_point_count_04m-05m | 4 m to 5 m |
| vegetation_point_count_05m-06m | 5 m to 6 m |
| vegetation_point_count_06m-07m | 6 m to 7 m |
| vegetation_point_count_07m-08m | 7 m to 8 m |
| vegetation_point_count_08m-09m | 8 m to 9 m |
| vegetation_point_count_09m-10m | 9 m to 10 m |
| vegetation_point_count_10m-11m | 10 m to 11 m |
| vegetation_point_count_11m-12m | 11 m to 12 m |
| vegetation_point_count_12m-13m | 12 m to 13 m |
| vegetation_point_count_13m-14m | 13 m to 14 m |
| vegetation_point_count_14m-15m | 14 m to 14 m |
| vegetation_point_count_15m-16m | 15 m to 16 m |
| vegetation_point_count_16m-17m | 16 m to 17 m |
| vegetation_point_count_17m-18m | 17 m to 18 m |
| vegetation_point_count_18m-19m | 18 m to 19 m |
| vegetation_point_count_19m-20m | 19 m to 20 m |
| vegetation_point_count_20m-25m | 20 m to 25 m |
| vegetation_point_count_25m-50m | 25 m to 50 m |

### 3.5.6 Vegetation proportions by height bin (vegetation_proportion_xxx)

The vegetation proportions by height bin are amongst the key parameters in the EcoDes-DK15 data set describing vegetation structure as they provide an indication of how the vegetation is distributed vertically within each cell of the raster. We calculated the proportions by dividing the vegetation count for each height bin (Table 4) by the total point count (total_point_count_-01m-50m) within a given 10 m x 10 m cell. Resulting proportions were multiplied by a factor of 10000, rounded to the nearest integer and converted to 16-bit integers. All calculations were done using *gdal_calc* based on the respective point count rasters (Sect. 3.3.5). The naming convention of the vegetation proportion descriptors "vegetation_proportion_xxx" follows the same convention as the vegetation point count descriptors (Table 4), whereby the suffix "xxx" is replaced with the respective height bin. Please note that height bins are spaced at 0.5 m intervals below 2 m and at 1 m intervals between 2 m and 20 m. Furthermore, the range above 20 m is split into only 2 bins: 20 m to 25 m and 25 m to 50 m.

Given the properties of the DHM/Point-cloud we recommend being cautious when interpreting differences in the lower height bins. It is likely that the inaccuracies in the point cloud complicate clear separation between points less than half a metre apart. Furthermore, note that the proportions in the 0 m - 0.5 m bin are likely biased towards an underrepresentation of the vegetation proportion in this height bin, due to challenges in separating vegetation from ground points during the pre-classification. Lastly, keep in mind that dense canopy layers in the upper story of the canopy will reduce penetration of the light beam to the lower canopy layers. This may result in few returns in the lower layers (for example Fig 3 d) even though perhaps vegetation is present in those layers.

### 3.5.7 Vegetation density or total vegetation proportion (vegetation_density)

Vegetation density is an important component of ecosystem structure. Here, we calculated the vegetation density as the ratio between the vegetation returns across all vertical height bins (vegetation_point_count_00m-50m) and the total point count (total_point_count_-01m-50m). Calculations were done using *gdal_calc* based on the two point count rasters (Sect. 3.3.5). Results were multiplied by 10000, rounded to the nearest integer and stored as 16-bit integers. In addition to actual difference between vegetation density in a cell, the vegetation_density descriptor is also influenced by the canopy properties, e.g. a dense upper layer will prevent penetration of the light beam to lower layers or even the ground, and the points sources within a cell, e.g. multiple sources from different viewing angles provide a more complete estimate of the vegetation density. These additional influences are important to keep in mind when interpreting the vegetation_density descriptor.

### 3.5.7 Canopy openness, or ground and water proportion (canopy_openness)

Canopy openness is an important ecological descriptor particularly of forest canopies, as it describes the amount of light penetrating through to the levels of the canopy. To some degree the canopy openness serves as the inverse for the vegetation density. For EcoDes-DK15, we calculated the canopy openness of a 10 m x 10 m cell as the proportion of the ground and water

points (ground_and_water_point_count_-01m-01m) to the total point count (total_point_count_-01m-50m) within the cell. The raster calculations were done using *gdal_calc*. Results were multiplied by 10000, rounded to the nearest integer and stored as 16-bit integers. Please note that the same considerations as for the vegetation_density descriptor (Sect. 3.3.7) regarding canopy properties and differences in point sources between the cells apply when interpreting the canopy_openness descriptor. In addition, it is important to note that building points will reduce the canopy openness the same way that vegetation points would.

**3.5.8 Building proportion (building_proportion)**

In a densely populated country such as Denmark, buildings form an important part of the landscape. For ecological studies the distance to buildings, their presence, absence or density may be of relevance. The building_propotion descriptor of EcoDes-DK15 provides a proxy for how much building infrastructure can be found within a 10 m cell. We calculated the descriptor as the number of building points (building_point_count_-01m-50m) divided by the total number of points (total_point_count_-01m-50m) within each cell using *gdal_calc*. Results were multiplied by 10000, rounded to the nearest integer and stored as 16-bit integers. While most returns from three dimensional infrastructure are classified as buildings in the DHM/Point-cloud, we would like to highlight that many roads are classified as ground (class 2) and some structures such as pylons and power lines were assigned a separate class (not described in (Geodatastyrelsen, 2015). These structures are therefore not included in the building_proportion descriptor. We would further like to note that the majority of building points are likely based on returns from the roofs of the buildings. Walls and other vertical structures are probably represented at a lower frequency in the point clouds. Finally, we would like to point the reader to the "DCE Basemap" (Levin, 2019) which may assist in the identification of basic land cover types that include buildings and other manmade structures.

**3.6 Auxiliary data**

In addition to the terrain and point cloud derived descriptors we provide three sets of auxiliary data with EcoDes-DK15. These are four layers of ALS point source information, a mask for inland water and a sea mask, as well as a shapefile of the footprints of the 1 km x 1 km tiles in the data set and their unique identifier.

**3.6.1 Point source information**

The point source attribute of the DHM/Point-cloud represents differences between sensor units or aircrafts that may have been used during the nationwide LiDAR campaign, differences in the acquisition time and date and differences in the viewpoint or acquisition angle of the cells. To aid in interpretation of descriptors that may be particularly influenced by point source, like the amplitude descriptors or the vegetation proportions, we provide summary information about the point sources within each 10 m x 10 m cell. We summarised this information in four descriptors, the "point_source_counts", "point_source_ids", "point_source_nids" and "point_source_proportions". For each tile (file name suffix = tile id), these descriptors are found in four subfolders bundled up in the parent "point_source_info" folder.

**point_source_ids** - Multi-layer raster containing one 16-bit integer layer for each point source id found in a tile. If a point with a given point source id is present the value of the cell is set to the point source id (an integer number) in the respective layer for the point source id, otherwise the value of a cell is set to 0. This multilayer raster can be used to match the file names of the point_source_counts and point_source_proportions rasters to a given point source id. Point source ids were extracted using *Opals Cell*.

**point_source_nids** - Single layer GeoTiff files containing the number of different point source ids in each cell stored as 16-bit integers. We calculated the number of point source ids based on the point_source_ids descriptor using *gdal_calc*.

**point_source_counts** - For each tile there are multiple rasters (up to four), one raster for each point source id found in the point cloud of the tile (see the *point_source_ids* descriptor). These rasters are named with an additional suffix, which matches the integer point source id for which the point counts are given in the raster (e.g. point_source_counts_xxxx_xxx_y*, where xxxx_xxx is the tile id and y* the integer point source id). The rasters contain the number of points per 10 m x 10 m cell for the respective point source id in the tile. Counts were extracted using the *OPALS Cell* module and stored as 16-bit integers.

**point_source_proportions** - For each tile there are multiple rasters (up to four), one raster for each point source id found in the point cloud of the tile (see the *point_source_ids* descriptor). These rasters are named with an additional suffix, which matches the integer point source id for which the point proportions are given in the raster (e.g. point_source_proportions_xxxx_xxx_y*, where xxxx_xxx is the tile id and y* the integer point source id). Each raster contains the proportion of the point counts for a given point source id in relation to the total point count per 10 m x 10 m cell. Calculations were carried out using *gdal_calc*. The final proportions were multiplied by a factor of 10000, rounded to the nearest integer and stored as 16-bit integers.

### 3.6.2 Water masks (inland_water_mask and sea_mask)

We also provide rasterized water masks for use cases that require masking inland water bodies or the sea. To represent all permanent lakes in Denmark, we merged three shapefiles containing (1) lakes protected by the Danish nature protection legislation (§3, available at https://arealinformation.miljoeportal.dk), (2) other valuable lakes (available on request at the Danish Farming Agency in the "good farming and environmental condition" data set) and (3) a layer containing the remaining rather small lakes and ponds (GeoDanmark, https://kortforsyningen.dk/). The combined shapefile is provided on the GitHub code repository (see below). We then burned the geometries within the shapefile into the 10 m x 10 m grid using *gdal_rasterize*. The masks are binary, a cell value of 1 indicates land and a value of -9999 (NoData) indicates sea or inland water, respectively. When using the masks please consider that the shape, presence and absence of water bodies and coastlines may fluctuate over time. We created the masks to present a snapshot of the water bodies as close as possible to the time point of the DHM/Point-cloud acquisition (spring 2014 - summer 2015), but inaccuracies may still arise. When combining the data with more recent observations, keep in mind that inland water bodies and coastlines may have changed since then. Finally, while we aimed to

produce the inland water mask to be as comprehensive as possible, some small ponds and water bodies may have been missed.
Note also that while some rivers are included in the sea mask, the inland water mask does not include rivers or streams. The
masks can be found in the "masks" subfolder of the complete data set.

### 3.6.3 Vegetation point date stamps (date_stamp_min, date_stamp_max, date_stamp_mode)

The time point at which the source data was collected may be of interest to certain applications that are using EcoDes-DK15
vegetation descriptors. These include for example, comparisons amongst regions where the data was collected under different
foliage conditions (leaf-on/leaf-off) or studies that require a precise timing of the sample such as change detection studies. To
better facilitate these applications, we generated three date_stamp descriptors that summarize the GPS time stamps of the
vegetation points within each 10 m x 10 m cell. The three descriptors are: date_stamp_min, date_stamp_max and
date_stamp_mode, which represent the earliest, latest and most common survey date for the vegetation points in any given cell
in the format "YYYYMMDD", where YYYY is the year in four digits, MM the month in two digits and DD the day in two
digits.
We used the *OPALS addInfo* module to generate a new "GPSDay" attribute for all vegetations points (classes 3,4,5) by dividing
the GPSTime (seconds since 6 January 1980) attribute by 86400 (seconds per day) and taking the floor value of the result. We
then exported the min, max and mode for each 10 m x 10 m cell using the *OPALS Cell* module, loaded the output rasters into
Python and converted the_GPSDay values into year, month and day in CET using the *datetime* module. Finally, we exported
the min, max and mode dates as 32-bit integers.
Note that the date_stamp descriptors only cover points that are classified as vegetation and therefore do not provide information
about the time point at which points belonging to other classes were surveyed (e.g., ground point, building points etc). We
chose to not include other point classes in the date_stamp descriptors, as we are aware that all versions of the source data sets
include some ground points from 2007, and as we believe that clear information about the vegetation points is most relevant
for the end-users conducting ecological research. Furthermore, determining the date_stamps was not possible for a proportion
of tiles where the GPSTime in the source data was not converted from seconds per GPS week to GPS time in seconds since 6
January 1980. A post-hoc conversion is not possible without the knowledge of the exact GPS week number, which is not
provided in the source data. In these cases, we assigned the NoData value to the date_stamps. The majority of the tiles affected
is located in the areas around Mols Bjerge and Sønderborg (Fig. 5). However, from auxiliary information about the source data
sets we know that these areas were surveyed April-May 2015 and October 2014, respectively.

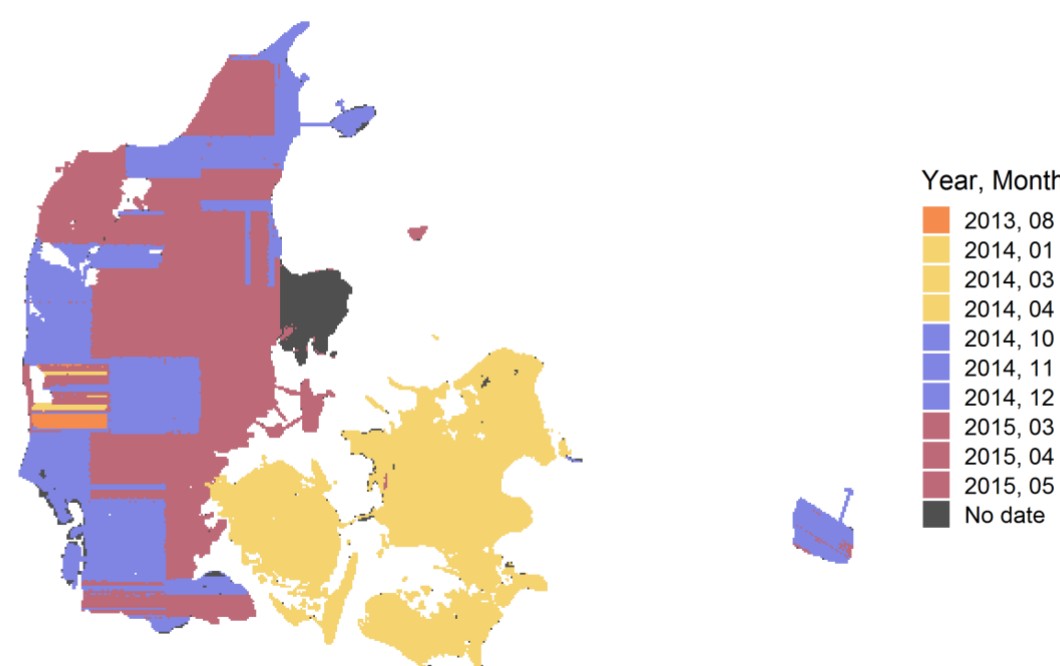

**EcoDes-DK15 Vegetation Point Collection Dates**

Aggregate of date_stamp_mode for each tile

Year, Month
- 2013, 08
- 2014, 01
- 2014, 03
- 2014, 04
- 2014, 10
- 2014, 11
- 2014, 12
- 2015, 03
- 2015, 04
- 2015, 05
- No date


**Figure 5:** Distribution of the most common survey date for the vegetation points in each tile of the EcoDes-DK15 dataset. The
data shown is aggregated for each tile from date_stamp_mode descriptor. The figure highlights that while the majority of the
vegetation points is from 2014/15, the data set also includes a small amount of vegetation points from 2013 in western Jutland.
Furthermore, surveys were conducted in all seasons, with vegetation points originating in spring, summer, autumn and winter.
Nonetheless, the majority of vegetation points comes from the leaf-off season. Lastly, the date_stamp descriptors could not be
derived for some regions as the GPSTime was not provided in the point clouds. However, from auxiliary information we know
that the surveys in the Mols Bjerge and Sønderborg areas were conducted in April-May 2015 and October 2014, respectively.
**3.6.4 Footprint file (tile_footprints.shp)**
To assist data access and creation of data subsets, we have produced an ESRI shapefile containing the footprints of all 1 km x
1 km tiles in the EcoDes-DK15 data set. The shapefile was generated based on the "dtm_10m" rasters and the tile identifier of
each footprint geometry is specified in the "tile_id" attribute column.

## 4. Data access and ecological use case example

### 4.1 Data access and handling

Depending on the extent of the study, it may be preferable to work with a subset of the data set rather than the nationwide VRT files (Fig. 6). We suggest starting by identifying the relevant EcoDes-DK15 descriptors of interest, then retrieving the relevant data from the repository and decompressing the archives (instructions provided on data repository). If the study area of interest covers a large fraction of Denmark's extent and sufficient processing power is available, the nationwide VRT data should provide the most convenient access to the selected descriptors. However, if the study area does not cover a large proportion of Denmark, then we suggest sub setting the data using the tile footprints to decrease demands on computational resources. After sub setting, local / regional VRT files or mosaics can be generated if needed. We provide an example R script illustrating how this sub setting could be done for the use case example shown in the next section on the code repository (manuscript/figure_7/subset_data_set.R). We have also made the resulting subset available as a "teaser" (5 MB) to help the reader assess the value of EcoDes-DK15 without having to commit to the multi-gigabyte download of the complete data set (see Sect. 6).

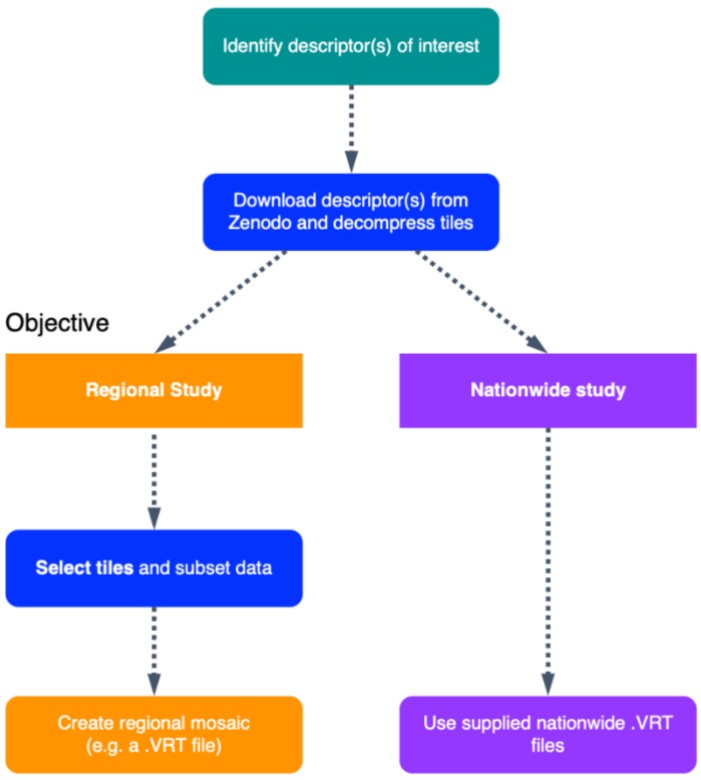

**Figure 6:** Schematic chart of two possible approaches for accessing and integrating EcoDes-DK15 data into ecological studies. The first step is to identify which descriptors are of interest, these descriptors can then be downloaded from the Zenodo repository and decompressed. Next a decision needs to be made whether the whole data set (nationwide) or only a subset of the tiles is required (e.g., a regional study). As the whole data set is relatively large (~94 GB), storage and processing limitations need to be taken into account when planning data processing and handling. If a subset of tiles is sufficient for a study, the provided tile footprints can be used to identify which tiles are required based on a geometry (e.g., a shapefile) of the study region(s). Finally, for easy data handling in subsequent analysis, a mosaic of the selected tiles can be created. For nationwide use we provided virtual mosaics (VRT files) containing all tiles for the descriptors. An R script illustrating how the sub setting can be done for a regional study can be found on the GitHub repository: https://github.com/jakobjassmann/ecodes-dk-lidar/blob/master/manuscript/figure_7/subset_dataset.R.

 **4.2 Use case example - ecological landscape stratification of Husby Klit nature protected area**

Figure 7 illustrates a use case for the EcoDes-DK15 data set with an example of an ecologically motivated landscape
stratification of the "Husby Klit" old-dune protected area in western Denmark. We developed this stratification for a group of
Master's projects carrying out vegetation monitoring in the area. Our aim was to capture the variation in the dominant
vegetation based on vegetation structure as well as the variation in fine-scale topography created by the dune systems across
the landscape. In addition to using the descriptors already provided, the stratification required us to derive a topographic
position index as well as grouping the point densities in height bins relevant to the characteristics of the three most common
dominant vegetation types (grass and heath, *Pinus mugo* Turra, *Pinus sylvestris* L.) in the area. The source code for this figure
contained in the code repository provides an example of how this can be achieved (manuscript/figure_7/figure_7.R).

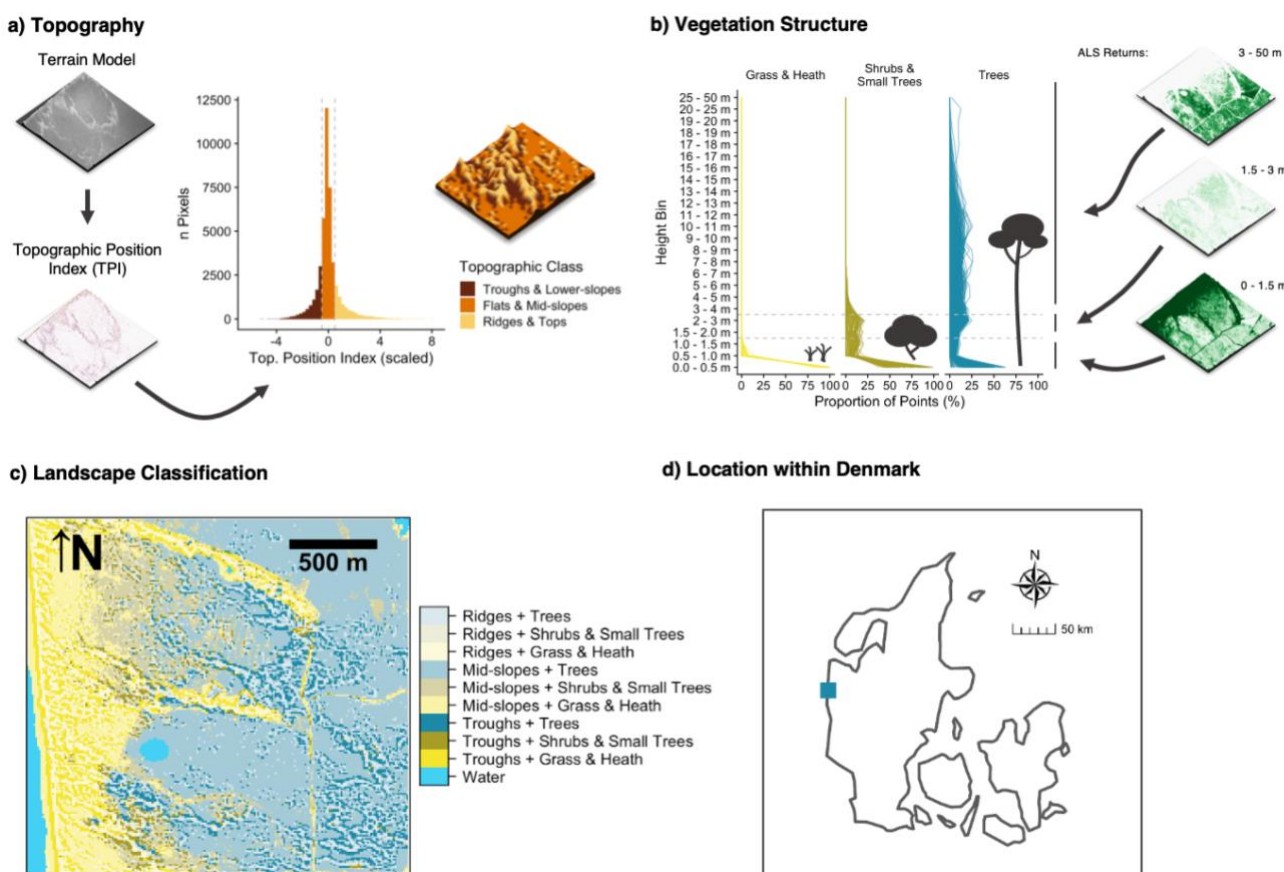


**Figure 7:** Use-case example: Landscape stratification of the Husby Klit protected area based on EcoDes-DK15 derived terrain
and vegetation structure descriptors. The target was to stratify the landscape of the Husby Klit "dune plantation" area in the
west of Denmark (56.2837 - 56.3024 °N, 8.1239 - 8.1600 °E) to facilitate stratified random sampling for vegetation monitoring.

We identified the four tiles overlapping with the boundaries of the protected area and derived a stratification based on two components: topographic position (a) and vegetation structure (b). We hypothesized that both components would influence the vegetation communities present. For the topographic position (a), we first derived and standardised the topographic position index (TPI) (Weiss, 2001) from the terrain model (dtm_10m). Following (Weiss, 2001) we then classified each cell based on the scaled TPI into three categories. A scaled TPI below a value of -0.5 was classified as a "trough or lower-slope", a scaled TPI between -0.5 and 0.5 as "mid-slope or flat", and a scaled TPI above 0.5 as a "ridge or top". For the vegetation structure component (b), we calculated the proportion of returns in three simplified height bins: 1) 0 m to 1.5 m, 2) 1.5 m to 3.0 m and 3) 3.0 m - 50 m. Here we included both ground and vegetation returns as the divisor for the standardisation, but not the returns from buildings or water. Based on *a priori* knowledge we deduced that there are three dominant vegetation communities within the protected area: communities dominated by grass and heath with vegetation growth generally below 1.5 m, communities dominated by shrubs and small trees (including the invasive *Pinus mugo*) with vegetation growth predominantly below 3.0 m, and communities dominated by trees (including the native *Pinus sylvestris*), generally with growth above 3.0 m. We used this knowledge to assign the three vegetation classes based on the proportion of point returns in the simplified height bins. For the "grass and heath" class we used a strict cut off with no points present above 1.5 m. For the "shrubs and small trees" class we used a fuzzy cut off allowing the proportion of points in the 3.0 m and above bin to reach up to 10% of the maximum proportion found in this heigh bin. All remaining cells were then assigned to the "trees" class. Finally, we combined the two classifications into one as illustrated in c). Panel d) shows the location of the protected area within Denmark. The 3D raster visualisations were generated using the rayshader v0.19.2 package in R (Morgan-Wall, 2020).

## 5. Discussion - limitations and future perspectives

Our data set demonstrates how the complex information in ALS point cloud data sets spanning more than 40.000 km$^2$, can be condensed into a compact data set of rasterized descriptors of interest for ecological studies. For the whole of Denmark, we provide 70 raster layers representing eighteen measures that describe a snapshot of vegetation height, structure and density, as well as topography and topography-derived habitat characteristics, including slope, aspect, solar radiation and wetness for the time period 2014-2015. These measures are of direct relevance for ecological research on species' habitat characteristics, distribution modelling, biodiversity and conservation applications. Condensing the ALS derived information into a compact set of raster descriptors makes it more accessible to the community of ecological researchers and practitioners, allowing them to access information on the vertical structure of vegetation and terrain otherwise difficult to obtain for large extents such as those of a whole country.

We would like to highlight some key ecological and physical limitations that should be kept in mind when using the data or derivatives. Firstly, we were able to only carry out a simple qualitative assessment of the errors in the EcoDes-DK14 data set within the scope of this project. All descriptors should therefore be seen as proxies for the geographical and biological

properties they describe. Errors in the original point cloud and DTM will have propagated through to the final descriptors and
future studies are needed to assess to which degree the proxy measures correlate with in-field data. Furthermore, the EcoDes
data set is a snapshot in time representing the state of the vegetation in the one and a half years between spring 2014 and
summer 2015 (with some exception in western Jutland, where the data is from 2013). Like anywhere on Earth, the landscapes
of Denmark may change over time and by the time point of publication of this data set over 5 years may have passed since the
collection of the source data. External data sources containing information about on-going or past changes (such as satellite
imagery - see below) might help overcome this bias. Additionally, the geographical differences in the timing of the point cloud
collection across the country (see Sect 6.3.4) may introduce noise and could affect cross-comparability of the data between
regions, for example due to seasonal differences in foliage (see e.g., Leiterer et al., 2015). Furthermore, there are implicit
limitations in spatial scale due to the set grain size of the data set. We chose a 10 m x 10 m grid for efficiency in computation
and data handling, as well as to overcome limitations in the density of the source point cloud (four to five points per m$^2$). Our
data set might therefore not serve well for capturing some ecological relevant variation in terrain and vegetation structures at
scales below the 10 m x 10 m grain size. We believe that our data set is nonetheless valuable in providing ecologically relevant
information at the geographical extent of Denmark.
While some of the descriptors in the presented data set such as elevation, slope and vegetation height are quite straightforward
to interpret, the ecological meaning of other descriptors – for example those related to vegetation structure – may not be as
obvious as they are influenced by multiple ecological and sensing methodology related factors. The *amplitude*, *point count*
and *point proportion* descriptors are amongst those measures. For example, while the (non-calibrated) amplitude in the
DHM/Point-cloud source data may generally relate to the reflectance properties of the surface that generated the return, the
incident light angle, scattering and subsequent generation of echoes may result in several different surfaces generating similar
amplitude signatures. Furthermore, the point counts may be influenced by a whole suite of factors, including incident light
angle, scattering, density of flight strips covering a given cell, as well as canopy properties - most importantly the penetration
ability. While standardising the point counts as proportions to the total counts may help to account for some of these factors,
it is likely that notable uncertainties will remain even in the proportions especially for lower layers of the canopy. Nonetheless,
we believe that these measures can be informative if appropriate care is taken in their interpretation.
Two code developments could enhance the EcoDes-DK15 processing workflow in efficiency and transferability: using gdal
Python bindings and switching to an open-source point cloud handler. First, for practical reasons we reverted to using gdal
binaries rather than the Python bindings as we encountered issues with the gdal bindings provided by the OPALS shell on our
computational server. Solving this issue and using the bindings instead of the binaries could reduce hard drive access time and
overheads from launching subprocesses and therefore potentially speed up the raster manipulations in the workflow. However,
as the point cloud processing takes the majority of time (we estimate 75-80%) we did not invest further resources to do so in
the first development round. Secondly, while our Python source code is open source and freely available, OPALS itself requires

the purchase of a software license, limiting the transferability of our code to projects which can afford the license. We did not explore alternatives to OPALS, but a redeveloped processing pipeline could make use of purely open source software benefiting from ongoing developments in the field, see for example the "Laserchicken" Python module (Meijer et al., 2020) and "lidR" R package (Roussel et al., 2020).

We believe that to realise the full potential of ALS derived data such as EcoDes-DK15 these data sets are ideally combined with other data sources including climate, field data and remote sensing observations. Climate data is especially relevant for addressing research on species-habitat relationships, distribution models and biodiversity studies and many studies have demonstrated the power of ALS observations in complementing climate data for such exercises (Coops et al., 2016; Zellweger et al., 2016). Like for other remote sensing products, field data is essential for validating inferences and putting biological meaning into ALS data (Coops et al., 2021) - this applies especially to the more complex structural vegetation measures in EcoDes-DK15. This could be achieved through field surveys combined with terrestrial and drone based ALS data, where the point density is much higher (e.g., Madsen et al., 2020). The potential benefits from fusing ALS data with other remote sensing products have been realised early on (Hyde et al., 2005) and demonstrated again since then (e.g., Coops et al., 2021; Montgomery et al., 2019; Manzanera et al., 2016). However, note that data fusion does not provide additional value in every use case (Xu et al., 2018; Ceballos et al., 2015; Boelman et al., 2016). We still believe that there is tremendous potential in combining EcoDes-DK15 with other types of remote sensing data. Fine-grain optical imagery could provide proxies for horizontal vegetation structure in grasslands where the vegetation is too small to be captured by the DHM/Point-cloud density (e.g., Malmstrom et al., 2017; Pazúr et al., 2021) and satellite derived time-series can provide unique temporal perspectives that describe parameters of seasonality (e.g., Boelman et al., 2016) and the historical context on disturbances and landcover change not captured in the single time-point ALS data (e.g., Senf et al., 2017; Pekel et al., 2016).

**6. Data availability**

The data is openly available under a Creative Commons by Attribution 4.0 license on Zenodo:

https://doi.org/10.5281/zenodo.4756556 (Assmann et al., 2021)

A small example subset "teaser" (5 MB) covering the 9 km x 9 km of the Husby Klit area (Fig. 7) is available on the GitHub code repository:

https://github.com/jakobjassmann/ecodes-dk-lidar/blob/master/manuscript/figure_7/EcoDes-DK15_teaser.zip

**7. Code availability**

The source code for the processing pipeline is openly available under a simplified BSD license via GitHub:

https://github.com/jakobjassmann/ecodes-dk-lidar

**8. Conclusions**

Open data sets like EcoDes-DK15 will allow ecologists with limited computational resources and little expertise in handling LiDAR point clouds to use large-scale ALS data for their research. We see our efforts not only as a first step for providing ready-to-use descriptors of local vegetation and terrain features, but also for providing an example workflow and tools that allow for the replication of the processing. We have described and documented the measures of terrain and vegetation structure contained in the data set and pointed out possible applications and limitations. We are confident that EcoDes-DK15 provides a meaningful collection of ecological descriptors at a 10 x 10 m resolution for the extent of a whole country and we encourage the community to use our workflow and collection of codes as inspiration to process other large-scale ALS data sets in a similar manner. Ultimately, we hope the publication of this data set will help facilitate the uptake of ALS-derived information by ecological researchers and practitioners in Denmark and beyond.

**9. Author contributions**

All co-authors developed the data set with focus on its ecological relevance, providing input on the ecological meaning, spatial scale and calculation of the descriptors. JJA developed the code with input from JEM. JJA carried out the computations. JJA led the writing of the manuscript, with all co-authors contributing to the manuscript in a collaborative manner. SN provided funding and supervision for this project.

**10. Competing interests**

The authors declare that they have no conflict of interest.

**11. Acknowledgements**

We would like to thank Andràs Zlinszky for his contributions to earlier versions of the data set, Charles Davison for feedback regarding data use and handling, as well as Matthew Barbee and Zsófia Koma for sharing their insights on the source data merger and Zsófia's script to generate summary statistics for the different versions of the DHM point clouds. Funding for this work was provided by the Carlsberg Foundation (Distinguished Associate Professor Fellowships) and Aarhus University

Research Foundation (AUFF-E-2015-FLS-8-73) to Signe Normand (SN). This work is a contribution to SustainScapes – Center for Sustainable Landscapes under Global Change (grant NNF20OC0059595 to SN).

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
