# Peer review of "EcoDes-DK15: High-resolution ecological descriptors of vegetation"

_Earth System Science Data, 2021_

## Author Comment (AC1)

**>> Vitezslav Moudry (Referee 1)**

>> First of all, I would like to congratulate the authors for their excellent work. I've been hoping for such an initiative for some time. You have done an excellent job. Please find below minor comments that could improve your dataset/manuscript.

Thank you very much for the positive appraisal of our work. We are very glad to hear our initiative was positively received and are grateful for the feedback and constructive criticism.

Line numbers in our replies refer to the updated version of the manuscript.

Referee comments are indicated with ">>".

**>> 1 Introduction**

>> I suggest to either simplify the introduction (by omitting some technical information) or provide better explanation, as in its current form it could be misleading (see my specific comments below).

This is an important point and we agree with the detailed issues raised in the comments below. We have adapted the text as suggested and outlined the specific changes made in response to the individual points below.

**>> Lines 43-45:** "*Some surfaces may scatter large parts of the light which may result in multiple returns of different strengths, vegetation is amongst those (Wagner et al., 2006).*" I am not a physicist (neither native speaker), but not sure with the terminology (wording) in this sentence. What do you exactly mean be "may scatter large parts of the light"? I would probably use reflect rather than scatter? In vegetated areas, the laser beams are usually reflected from several layers of vegetation. The interaction of the laser beam with the canopy is then characterized by multiple returns from several depths of the vegetation. The first return typically comes from the vegetation canopy surface and the second and possibly other returns follow (intermediate returns from leaves and branches), with the last one ideally being a return from the terrain. In other words, LiDAR pulses can penetrate through gaps in the vegetation canopies and register multiple returns representing both above-ground objects and the terrain.

Yes, this is indeed not clear. We have replaced the sentence with the following two sentences that hopefully summarise things with more clarity:

**Line 41-45:** If an object intercepting the light pulse is smaller than the beam's footprint (e.g., a leaf or a branch of a tree) only a part of the light pulse will be reflected from that object while other objects further along its path might reflect the rest. A single light pulse therefore often generates two or even more returns, allowing - to some degree - for the penetration of forest canopies (Ackermann, 1999).

Ackermann, F.: Airborne laser scanning—present status and future expectations, ISPRS J PHOTOGRAMM, 54, 64–67, https://doi.org/10.1016/S0924-2716(99)00009-X, 1999.

We also updated the next sentence adding the "return number" to the list:

**Line 45-47:** Often, the raw signal is processed by the survey provider and the resulting data is delivered to the end user in the form of a point cloud of discrete returns, where each point is associated with information on geographic location, return strength (amplitude), return number, acquisition timing etc. (Vo et al., 2016).

**>> Lines 49-50:** What do you mean by ALS methodology. That is quite vague. Do you mean data acquisition, or point clouds processing?

As the preceding paragraph mainly deals with the data acquisition process, we think the term is best fitting here. We have changed the sentence accordingly, it now reads:

**Line 50-51:** For further information on ALS data acquisition, we recommend Vo et al. (2016), Vierling et al. (2008) and Wagner et al. (2006).

**>> Lines 50 - 52:** Maybe add one or two sentences about data filtering/classification. So those unfamiliar with LiDAR point clouds get the idea about how the point clouds are classified/filtered. You refer to classification later in the text. You write: *„Depending on the properties of the LiDAR sensor, different surfaces will reflect the light with different strengths, allowing, for example, to separate returns from vegetation from those of bare- ground"*, but returns intensity is not the main characteristic that allows to separate vegetation from ground. This may give to an unexperienced user feeling that laser pulse is able to recognize building from vegetation, which is not true (see Sithole and Vosselman, 2004; Moudrý et al. 2020). Most of the state-of-the-art filtering techniques uses the relative positions of the points (similar situation is when distinguishing buildings from vegetation). Use of additional parameters, such as those tested in Wager et al. 2006 is to my knowledge rather experimental.

>> Sithole, G. Vosselman, Experimental comparison of filter algorithms for bare-Earth extraction from airborne laser scanning point clouds, ISPRS J. Photogramm. Remote Sens. 59 (2004) 85–101,

>> Moudrý, V., KlápsᴛÄ⟩, P., Fogl, M., Gdulová, K., Barták, V., & Urban, R. (2020). Assessment of LiDAR ground filtering algorithms for determining ground surface of non-natural terrain overgrown with forest and steppe vegetation. Measurement, 150, 107047.

Thank you for pointing this out. We agree there is a risk for confusion here and we thus have added two sentences to clarify the issue. The beginning of the paragraph no reads:

**Lines 52-56:** Based on point position and neighbourhood context it is possible to separate ground and vegetation returns in ALS point clouds, allowing for the calculation of descriptors of terrain and vegetation structure. Filtering bare ground from the point cloud may be achieved with algorithms (Moudrý et al., 2020; Sithole and Vosselman, 2004), while more complex segmentation of the point clouds into object classes (such as vegetation, buildings, etc.) is done manually or with the help of supervised machine learning (see Lin et al., 2020 for a recent overview).

Lin, Y., Vosselman, G., Cao, Y., and Yang, M. Y.: Active and incremental learning for semantic ALS point cloud segmentation, ISPRS J PHOTOGRAMM, 169, 73–92, https://doi.org/10.1016/j.isprsjprs.2020.09.003, 2020.

Moudrý, V., Klápště, P., Fogl, M., Gdulová, K., Barták, V., and Urban, R.: Assessment of LiDAR ground filtering algorithms for determining ground surface of non-natural terrain overgrown with forest and steppe vegetation, MEASUREMENT, 150, 107047, https://doi.org/10.1016/j.measurement.2019.107047, 2020.

Sithole, G. and Vosselman, G.: Experimental comparison of filter algorithms for bare-Earth extraction from airborne laser scanning point clouds, ISPRS J PHOTOGRAMM, 59, 85–101, https://doi.org/10.1016/j.isprsjprs.2004.05.004, 2004.

**>> Lines 52 – 53:** *"While early applications for ALS were focused on generating simple digital elevation models (DEMs) for city and landscape planning, as well as canopy height estimates for commercial forestry,..."*. I thing that the references used: Bakx et al., 2019; Vo et al., 2016 are not relevant for landscape planning or forestry. You might

consider Fogl and Moudrý (2016) for city planning and Nilsson et al. (2017) for forestry (but in fact these are quite recent rather than early applications).

>> Fogl, Michal, and VítA̾zslav Moudrý. "Influence of vegetation canopies on solar potential in urban environments." Applied Geography 66 (2016): 73-80.

>> Nilsson, M.; Nordkvist, K.; Jonzén, J.; Lindgren, N.; Axensten, P.; Wallerman, J.; Egberth, M.; Larsson, S.; Nilsson, L.; Eriksson, J.; et al. A Nationwide Forest Attribute Map of Sweden Predicted Using Airborne Laser Scanning Data and Field Data from the National Forest Inventory. Remote Sens. Environ. 2017, 194, 447–454.

Indeed this was inappropriate referencing from our end. Instead we suggest to cite Ackermann (1999). We have modified the sentence to also accommodate for the newly added sentences at the beginning of the paragraph and to address the next point of concern raised. It now reads:

**Line 56-58:** Early applications for ALS were focussed on generating simple digital elevation models (DEMs), city and landscape planning, as well as forestry (Ackermann, 1999), but over the last decades applications have expanded into other fields, including amongst others the calculation of terrain and vegetation measures for ecological research.

Ackermann, F.: Airborne laser scanning—present status and future expectations, ISPRS J PHOTOGRAMM, 54, 64–67, https://doi.org/10.1016/S0924-2716(99)00009-X, 1999.

**>> Line 54:** Exactly which recent advances do you have in mind? Sure, the technology matured and it is now easier to acquire point clouds of high density, especially for large areas, but not sure if we were limited in calculation of complex measures in the past? Besides, after this information you continue with terrain derived measures of ecological interest, such as slope, but these are not complex measures and can be calculated using point clouds of comparatively low density (especially if you use 10m resolution).

This is a valid point. We have changed the sentence to be more descriptive of the historical developments, rather than attempting to simplify a reason why. See also the response to the previous point raised. The sentence now reads:

**Line 56-58:** Early applications for ALS were focussed on generating simple digital elevation models (DEMs), city and landscape planning, as well as forestry (Ackermann, 1999), but over the last decades applications have expanded into other fields, including amongst others the calculation of terrain and vegetation measures for ecological research.

Ackermann, F.: Airborne laser scanning—present status and future expectations, ISPRS J PHOTOGRAMM, 54, 64–67, https://doi.org/10.1016/S0924-2716(99)00009-X, 1999.

**>> Lines 58 – 61:** Not sure here if having sensor and point cloud characteristics in one sentence makes sense. I would simplify it and use solely the point cloud characteristics as it will depend on the used sensor. Besides, it is not only sensor (wavelength, footprint, etc.) but also surface reflectivity what matters. Not sure, that the review by Bakx et al. (2019) is the best citation here.

We are happy to go ahead with the simplification and have removed "sensor characteristics" from the sentence as suggested. It now reads:

**Line 62-63:** It is important to note that point cloud characteristics may limit the type of measures that can be meaningfully derived from ALS data (Bakx et al., 2019).

However, we believe that Bakx et al. (2019) is a good citation here to support the statement. From our perspective, the discussion of the influence of point cloud

characteristics on the calculation of vegetation metrics in the paper provides a good high-level overview on the subject (see especially p.1055 second column first paragraph). Although we are very open to suggestions for additional references that might be relevant here.

**>> 2 Source data and processing workflow overview**

**>> Line 104**: Nord-Larsen et al (2017) -> Nord-Larsen et al. (2017).

Corrected.

**>> Line 106:** S is missing; ETRS89 UTM32N; No EPSG code for DVR90? Is it EPSG: 5799?

We see, that our description was unclear and changed the sentence as follows:

**Line 114-116:** The DHM/Point-cloud product is provided in LAZ-format and in the compound coordinate system for Denmark (ETRS89 / UTM zone 32N + DVR90 height - EPSG:7416).

For additional clarity, we added the vertical reference system to the section outlining the dtm_10m descriptor:

**Line 225:** Values represent the elevation above sea level in metres (DVR90, EPSG: 5799) [..]

**>> Lines 113 - 116:** Not sure I completely understand this? Does it mean that complementary data (i.e. terrain or point cloud) are not available at all or just that tiles are somehow shifted? If this is the case, it would make sense to resample/re-tile them. It is 291 km squares which might be missing for potential users (0.7% of Denmark).

There is no shift in the extent / geolocation of the tiles between the two data sets. The 1 km x 1 km tiling of Denmark is identical for both. However, both datasets are not fully complementary in their coverage. A small number of tiles that are present in the DTM are not available in the point cloud dataset and vice versa. We had previously investigated why this could be the case and found the missing tiles that we inspected were located in the sea or water bodies. We suppose that the commercial provider processing the point cloud and DTM for the Danish authorities (two different provider), may have dropped these tiles from processing as these are empty of land points / only contain sub-tidal areas. As the 291 affected tiles only form 0.58% of the total tiles, we did not investigate any further.

We have adapted the text to improve the clarity of the statement made and updated the relevant numbers based on the new version of the source data set (see reviewer 2 comments):

**Line 126-128:** The 1 km x 1 km tiling of the DHM/Terrain 2014/2015 and DHM/Point-cloud data sets 2014/2015 match in extent and geolocation. However, a small number of tiles (n = 30) in the DHM/Point-cloud data sets did not have corresponding tiles in the DHM/Terrain data sets, these were removed prior processing resulting in the total of 49673 tiles shown in Table 1.

**>> Lines 142 - 143:** I suggest to write datasets or inputs instead of variables, which you use for the characteristics derived from the point cloud and DTM. Actually, you use term descriptors in the Title, Introduction, Conclusions, and some figures and tables while in chapters 2, 3, 4, and 5 you use term variables.

We changed "variables" to "data sets" in this sentence as suggested. It now reads:

**Line 164-165:** Together the Python scripts and modules allow to generate the ecological descriptor outputs from the two input data sets.

We also changed the term "variable" to "descriptor" throughout the manuscript, tables and figures for consistency.

**>> 3 Data set description and known limitations Line 149:** S is missing; ETRS89 UTM32N

Corrected.

**>> 3.4. Elevation-model derived variables**

**>> Line 194:** Kopecký et al (2020). -> Kopecký et al. (2020).

Corrected.

**>> Aspect and Slope:** Equally relevant would be to derive slope at 0.4 m grain size and than average it into the 10 m grain (although more computationally demanding). This way actually better preserves the original values at finest resolution, which might be important for some users. Maybe you want to mention it (see Grohmann, 2015 and Moudrý et al. 2020) similarly as you mention limitations of TWI.

>> Grohmann, C. H. (2015). Effects of spatial resolution on slope and aspect derivation for regional-scale analysis. Computers & Geosciences, 77, 111-117.

>> Moudrý, V., Lecours, V., Malavasi, M., Misiuk, B., Gábor, L., Gdulová, K., Símová, P. and Wild, J., 2019. Potential pitfalls in rescaling digital terrain model-derived attributes for ecological studies. Ecological Informatics, 54, p.100987.

We actually did derive the slope and aspect descriptors like this for a previous version of the data set. However, for consistency with the scale at which all the other derived descriptors are calculated we decided to switch to aggregating first and then deriving the aspect and slope parameters. We see that it is important to point out these considerations to the user of the data set and have added the following sentence to the end of the paragraphs describing the aspect and slope descriptors.

**Line 240-243:** Please also note that we calculated the aspect descriptor from the 10 m aggregate of the DTM/Terrain data set rather than deriving it from the 0.4 m original resolution rasters and then aggregating it. The latter approach could represent the aspect/slope at the original resolution better (Grohmann, 2015; Moudrý et al., 2019), but would create inconsistencies within how the remaining DTM/Terrain descriptors are calculated in this dataset.

**Line 256-259:** Please also note that we calculated the slope descriptor from the 10 m aggregate of the DTM/Terrain data set rather than deriving it from the 0.4 m original resolution rasters and then aggregating it. The latter approach could represent the aspect/slope at the original resolution better (Grohmann, 2015; Moudrý et al., 2019), but would create inconsistencies within how the remaining DTM/Terrain descriptors are calculated in this dataset.

**>> Solar Radiation:** Is it reasonable to calculate solar radiation in **MJ/cm2/yr** when you are using 10m resolution? You do not even have centimeter resolution in original DTM. I mean "average ecologist" can easily overlook this fact and considerably underestimate the solar potential in individual cells. Besides, if you calculate the solar potential as MJ/yr per cell, you will not have to use the conversion factor of 1000. In addition, I had to look to original text by (McCune and Keon, 2002) to realize that "ln" is natural logarithm (first I thought that it is a unit I am not familiar with) and that the data are not in arithmetic scale. So, if I am correct, to get the actual solar radiation in

individual cell I have to multiply by 10 000 (cm -> 10m), divide by 1000 (conversion factor) and convert to arithmetic scale (if needed), right?

Thanks a lot for pointing this out. We have now calculated the solar radiation in Mega Joule per year per 100m$^2$, where 100 m$^2$ (= 10^6 cm$^2$) represents the area of a 10 m x 10 m cell. We updated the text, formula and the relevant row in Table 1 accordingly:

**Table 1** row "solar_radiation":

"Unit" changed to: MJ x 100$^{-1}$ m$^{-2}$ x yr$^{-1}$
"Format" changed to: 32-bit integer
"Conversion factor" changed to: 1

**Line 299-306:** We estimated the amount of incident solar radiation received per cell (100 m$^2$) per year from the slope and aspect computed as described above. Calculations were implemented using gdal_calc, following equation 3 specified in McCune and Keon (2002):

solar_radiation = 10^6×e^(0.339 + 0.808×cos(L)×cos(S) - 0.196 ×sin(L)×sin(S) - 0.482×cos(180 -|(180 - A)|)×sin(S))

where L is the centre latitude of the cell in degrees, S is the slope of the cell in degrees and A is the aspect of the cell in degrees. The resulting estimate is given in: MJ x 100-1 m-2 x yr-1 (McCune and Keon, 2002). Slope and aspect for each 10 m x 10 m grid cell were sourced from the slope and aspect rasters. We saved the result as 32-bit integers.

We also update the example of the solar_radiaiton variable in Figure 2 and associated caption to reflect the changes:

[Figure]

**Figure 2:** Illustration of the terrain model derived descriptors for a 1 km x 1 km tile in the Mols Bjerge area (tile id: 6230_595). An orthophoto and the tile location relative to Denmark are shown in (a). The terrain model (dtm_10m) is illustrated in (b). The terrain derived descriptors comprise of: c) the topographic aspect, d) the topographic slope, e) the heat load index following Kuehne et al. f) the estimated incident solar radiation, g) the landscape openness mean, h) the landscape openness difference in the eight cardinal directions and i) the topographic wetness index (TWI) based on Kopecký et al. (2020). For visualisation purposes, we amplified the altitude above sea-level by a factor of two in the 3D visualisations and divided the solar radiation values by 10$^5$. The 3D raster visualisations were generated using the rayshader v0.19.2 package in R

(Morgan-Wall, 2020). Orthophoto provided by the Danish Agency for Data Supply and Efficiency (https://sdfe.dk/hent-data/fotos-og-geodanmark-data/).

>> Besides, I strongly suggest that in a next version of this dataset you calculate solar radiation monthly and to use some better model for estimation (e.g. r.sun implemented in GRASS; Súri & Hofierka, 2004). The model you used does not account for cloud cover, so you assume clear sky all the year. And it does not account to shading due to adjacent topography, which is a shame, considering how accurate terrain data you are using (although I realize that Denmark is relatively flat). Consequently, the first sentence of the paragraph "*Incident solar radiation is a key parameter for plant growth and indicator for local microclimate.*" is in my opinion irrelevant for your dataset as you do not consider local (microclimate) terrain conditions in your calculations. Maybe you should mention this in the text (or documentation). I would not count that users of your data will look at the original article you cite.

>> Súri, M., & Hofierka, J. (2004). A new GISâ based solar radiation model and its application to photovoltaic assessments. Transactions in GIS, 8(2), 175-190.

We agree that accounting for cloud cover and shading when calculating the solar radiation descriptor will be a worthwhile addition to the next version of the dataset. However, this will require integration of additional data, such as cloud cover maps, which was beyond our intent for the current dataset. Here, the measure we aimed to calculate was the potential solar radiation at the resolution of the data set, i.e., 10 m x 10 m solely based on the source ALS data. Light is important for plant growth but we believe that in Denmark light availability is rather controlled by shadowing from other plants rather than from terrain. Like you mention, Denmark is a flat country where shadow effects at the scale of 10 m likely have relatively minor impact and mostly outside the growing season when the solar angle is lower. Therefore, the most notable influence of incident sunlight at the scale of our data set is likely the warming effect on the local climate, thereby impacting water dynamics and plant growth. Something we also observed in previous work (Moeslund et al. 2019 cited in the manuscript). We agree that "microclimate" does imply smaller scales than what we intended here. We have removed the reference to microclimate in the first sentence of the paragraph and provided some additional explanation on the descriptor.

**Line 295-301:** Incident solar radiation is a key parameter for plant growth as it represents the electromagnetic energy available to plants required for photosynthesis. However, in the comparatively flat country of Denmark, shading by other vegetation likely exerts a larger influence on photosynthetic activity than terrain related shading. Here, the impact of incident solar radiation on the local climate likely plays a more important role for determining plant growth due to its influence on drought/water dynamics (Moeslund et al., 2019).

**>> Line 286:** twi (also in Table 2); Line 287: TWI; be consistent.

In the case of the title of the subsection and the table, "twi" refers to the name of the descriptor. We decided to declare all descriptor names and associated folder & file names using lower case letters. Keeping the "twi" lowercase in these two instances is therefore done for consistency within the naming conventions of the dataset. We appreciate that this breaks with the convention of capitalising abbreviation and the common use of "TWI" as an acronym in the literature, but believe that in this case this is the way to go for internal consistency. However, we spotted a typo in the first sentence of the paragraph (topography spelled with a capital T, line 287 of original manuscript), which we have now corrected.

**>> 3.5 Point-cloud derived variables**

**>> Lines 435 - 436:** The way I understand the term „texture", it is just other way (than vegetation density) to describe vegetation structure. So why should I worry about it, it is a property that this variable (vegetation density) should describe. Or did I misunderstand what you mean? Besides, do you have any reference that would support

your statement that different canopies scatter light differently, causing lower or higher number of returns? Is it that significant difference that it is worthy to mention?

We agree with the Reviewer's comment that this sentence was not clear and mentioning texture in that context is redundant. We therefore skipped the "texture" part of this sentence which now reads:

**Line 480-483:** In addition to actual difference between vegetation density in a cell, the vegetation_density descriptor is also influenced by the canopy properties, e.g. a dense upper layer will prevent penetration of the light beam to lower layers or even the ground, and the points sources within a cell, e.g. multiple sources from different viewing angles provide a more complete estimate of the vegetation density.

**>> Lines 449-460:** Building proportion: Based on some random tile (which included only small buildings) I guess that what is classified as buildings are typically returns from the roofs (or are there also vertical walls represented)? Not sure, but maybe this should be also mentioned.

The classification of the point cloud was carried out by the Danish authorities (SDFE/Kortforsyning) or their commercial providers, as described in Sect. 2.2 and 3.5. Unfortunately, no public documentation of the exact classification method is provided and we therefore can not provide a detailed description on how each class was assigned. However, from our experience working with the point cloud we know that some parts of building walls were classified as "class 6 buildings". See screenshot below for examples. Nonetheless, we added a sentence highlighting to the building proportion paragraph to highlight the issue to the reader.

**Lines 504-506:** We would further like to note that the majority of building points are likely based on returns from the roofs of the buildings. Walls and other vertical structures are probably represented at a lower frequency in the point clouds.

Revisiting the paragraph, we also noticed that the use of the adjective "nearest" in relation to the buildings was unclear in the second sentence of the paragraph. We have therefore removed the adjective to simplify the sentence, it now reads:

**Line 496-497:** For ecological studies the distance to buildings, their presence, absence or density may be of relevance.

[Figure]

Screenshot showing settlement in Vejle Fjord tile 6171_541. Points classed as "buildings" show in red, all other points in light grey.

**>> Yifang Shi (Referee 2)**

>> The authors demonstrated a great work of generating country-wide ALS-derived products for ecological applications in Denmark. It is a huge effort to handle a huge ALS dataset and develop the workflow. With the availability of country-wide ALS datasets, this type of work will be of great interest to researchers and policymakers all over the world.

It is great to read that our work is of interest! Thank you very much for the helpful feedback provided below.

Line numbers in our answers refer to the lines in the revised manuscript unless otherwise stated.

Referee comments are indicated with ">>".

>> My questions and comments are as follows,

>> Regarding the input data, the authors mentioned they used ALS point cloud and DTM at 0.4m resolution. Are these two datasets required? What if the resolution of DTM is courser (e.g. 1m or 5m) or there is no DTM product available? How can the user adjust the workflow?

This is an excellent point to raise. Yes, a point cloud and paired DTM raster are required for the processing. However, we provide a script that can generate a missing DTM if only a point cloud is available, also we did not use it when generating the EcoDes-DK15 data set. Furthermore, with a minor change in the code, the processing workflow can be adapted to aggregate a DTM of any grain size that is a clean divisor of 10 m (or any other desired output grain size). All that said, it is important to note that the generation of a DTM requires the point cloud to be pre-classified following the ASPRS LAZ 1.1-1.4

standard. Unclassified point clouds are not able to be processed with our workflow without substantial pre-processing steps or more detailed changes to the code.

We have added the following text to the manuscript to make this clear and to provide guidance should the reader be interested in this:

**Line 110-111:** Point classification follows the ASPRS LAS 1.3 standard (ASPRS, 2011), including for example ground, vegetation, and buildings.

**Line 147-152:** We generated the processing workflow so that it should be possible to adapt it to other point cloud data sets. However, the effort required in achieving this will vary depending on various features of the point cloud data set in question (such as tile naming convention, resolution etc.). A key requirement for a fairly straightforward conversion of the workflow is that the point cloud is pre-classified, ideally following the ASPRS LAS 1.1-1.4 standards (ASPRS, 2019). We provide a helper script that can be used to generate a raster DTM from the point cloud should this not be available, see the documentation on the GitHub repository for the details.

**Line 353-354:** The DHM/Point-cloud point cloud was pre-classified into eleven point categories (Geodatastyrelsen, 2015) following the ASPRS LAS 1.3 standard (ASPRS, 2011).

>> How accurate is the pre-classification of the raw ALS point cloud? For example, do the vegetation points contain any powerlines, buildings, etc? How did you remove those noises?

The point cloud was pre-classified by the commercial provider to the Danish government authorities who commissioned the collection of the point cloud. The classification has been carried out according to the ASPRS LAS 1.3 standard (ASPRS 2011), which separates vegetation points from buildings, power lines and bridges. Unfortunately, publicly available data on the accuracy of the classification are limited at the moment. The Danish authorities ran an in-detail quality check upon receival of the point cloud from the commercial provider (Flatman et al. 2016) and carried out some manual corrections (Flatman et al. 2016), but few statistics are provided on the overall accuracy of the classification.

Anecdotally, based on our personal experience handling the point clouds, the classification seems to separate vegetation from building and powerlines very well. No cases of power lines being miss-classified are known to us. In rare cases we noticed very tall structures being classified as buildings (something also mentioned in Flatman et al. 2016). We cannot comment on vegetation being misclassified as buildings without further in-depth assessment. Tall vegetation (class 5) was used as a catch all class during the classification (Flatman et al. 2016) and some noise will therefore be contained in that class.

To reduce the overall noise as much as possible, we excluded all points classified as power lines and bridges in the calculated descriptors. Building points were only included for certain descriptors, such as the building point counts and proportions as outlined in the manuscript. Finally, for the vegetation point derived descriptors we excluded all points with a normalised height exceeding 50 m above ground on the assumption that trees in Denmark are unlikely to exceed that height.

A systematic, independent evaluation of the classification would be desirable, but would burst the scope of our current project and exceed the resources we have available. However, we recognise the importance of being more clear about the limitations of the classifications and have added the following sentences to the manuscript to inform the reader:

**Line 358-363:** The overall classification accuracy of the point cloud was assessed by the Danish authorities (Flatman et al., 2016), but limited information is available for the accuracy in each class. Thus, some degree of noise should be assumed across all classes. The tall vegetation category (class 6) was used as a catch-all class if classification failed, as often the case for very tall buildings and structures (Flatman et al., 2016). To reduce the noise related to such structures, we removed vegetation points with a normalised height exceeding 50 m above ground when calculating the vegetation point counts.

We also updated the sentence describing the mis-classification of a telecommunication tower south of Aarhus in Section 3.5.3 to remove ambiguous inference of vegetation growth on the tower, it now reads:

**Line 422-423:** For example: A tall communications tower can be found just south of Aarhus and returns from the tower were miss-classified as vegetation.

ASPRS: LAS Specification Version 1.3 – R11, American Society for Photogrammetry & Remote Sensing, Bethesda, Maryland, 2011.

Flatman, A., Rosenkranz, B., Evers, K., Bartels, P., Kokkendorff, S., Knudsen, T., and Nielsen, T.: Quality assessment report to the Danish Elevation Model (DK-DEM), Agency for Data Supply and Efficiency, Copenhagen, Denmark, 2016.

>> Did you also include the buildings in point cloud processing in section 3.5 (lines 318-319)?

Yes, we did. Unfortunately, we forgot to list them here. Thank you for highlighting this issue. Detailed handling of the building points can be found in the descriptions of the point_count and building_proportion descriptors (Sec. 3.6.5 and 3.6.8). We have corrected the mistake of omitting the building points in the leading paragraph of Section 3.5, it now reads:

**Line 354-356:** For the EcoDes-DK15 data set, we restricted the analysis to six of these classes, including ground points ("Terræn") - class 2, water points ("Vand") - class 9, building points ("Bygninger") – class 6, as well as low ("lav"), medium ("mellemhøj") and high vegetation ("høj vegetation") - classes 3, 4 and 5, respectively.

>> Can you specify the limitations of the EcoDes-DK15 dataset you mentioned in Figure 3 line 332?

Yes, we are very glad to do so. We have added the following sentences to the end of the figure caption:

**Line 381-387:** [...] Please note how the classification of the point cloud classification does not separate between very low growing vegetation (e.g., grass) and ground points in the agricultural field shown in a), and how returns from water are only registered in shallow areas close to the water bodies edge, such as exemplified by the forest pond in b). Lastly, we would like to point the reader to the general limitations of ALS in penetrating forest canopies such as those shown in c) and d). While the upper layers of the canopies are well resolved in both cases, the laser scanning struggles to capture some aspects of the lower layers; the ground returns were frequently blocked by the thick canopy in c) and the laser fails to meaningfully characterise understory vegetation and stems in d).

>> Can the employed workflow generate new user-defined variables?

Yes, the workflow is completely modular. User-generated routines to calculate new descriptors can be easily included and already defined descriptors can be excluded as desired. We have added a sentence to highlight this to the reader.

**Line 152-153:** Finally, the modular nature of the processing workflow allows for only a subset of the output descriptors to be calculated and the integration of additional processing routines for any new user-defined descriptors.

>> In the point cloud processing, did you only use the XYZ coordinates information from the ALS point cloud? Have you considered the usage of intensity value since it can be very relevant for vegetation morphological and biochemical properties?

The ALS survey outputs were recorded as full waveform then processed to distinct returns by the commercial provider tasked with generating the point cloud by the Danish authorities. The intensity value ("amplitude") of each distinct return is stored as an attribute in the point cloud. The latter forms the base of the two amplitude descriptors as outlined in the manuscript (Sect. 3.5.1).

>> Have you evaluated the accuracy of the generated products? Giving an overview of the final products would be very useful. For example, where the error usually occurs? Are there still noises that remain? Which variables can be directly used and which ones need further processing for a better interpretation.

We did not carry out a quantitative evaluation of the generated products against ground-validation data and are therefore not able to make statements about the quantitative nature of the error associated with our derived descriptors. However, the physical accuracy of the point cloud was assessed by the Danish authorities (see manuscript Section 2.2, Geodatastyrelsen 2015 - in Danish, as well as Flatman et al. 2016, and Thers et al. 2017 p.104-105 for a more detailed summary in English). On our side and for this study, we carried out extensive qualitative assessments of our output descriptors. Much of the discussion regarding the ecological limitations in paragraphs containing the descriptor descriptions (Sect. 3.4 and 3.5), as well as the two paragraphs dedicated to discussing the data set's limitations in the discussion section (Sect. 5, paragraphs 2 & 3) stem from this qualitative analysis of the products.

We believe that a full quantitative evaluation of the product's outputs would be a meaningful undertaking, but at the same time it would unfortunately extend beyond the scope of the presented project. Such an undertaking would require, in many cases, the collection of new data in the field and perhaps even the development of novel methods to meaningfully compare those descriptors that do not have an equivalent that can be readily measured in the field (such as the amplitude, terrain openness and point proportion descriptors). All that said, we believe that our manuscript would benefit from making this clear to the reader and have added the following sentences to the second paragraph in the discussion section.

**Line 659-662:** [...] Firstly, we were able to only carry out a simple qualitative assessment of the errors in the EcoDes-DK14 data set within the scope of this project. All descriptors should therefore be seen as proxies for the geographical and biological properties they describe. Errors in the original point cloud and DTM will have propagated through to the final descriptors and future studies are needed to assess to which degree the proxy measures correlate with in-field data. Furthermore, [...]

And updated the sentence listing information on the data in Sect. 2.2 to include a reference to Flatman et al. 2016:

**Line 113-114:** Additional information on the data sets can be found in Geodatastyrelsen (Geodatastyrelsen, 2015 - in Danish), Thers et al. (2017), Nord-Larsen et al. (2017) and in the quality assessment report by Flatman et al. (2016).

Geodatastyrelsen: Dataspecifikation for Danmarks Højdemodel Punktsky. Data version 2.0 - Januar 2015., Geodatastyrelsen, Copenhagen, 2015. https://kortforsyningen.dk/sites/default/files/old_gst/DOKUMENTATION/Data/dk_dhm_punktsky_v2_jan_2015.pdf

Flatman, A., Rosenkranz, B., Evers, K., Bartels, P., Kokkendorff, S., Knudsen, T., and Nielsen, T.: Quality assessment report to the Danish Elevation Model (DK-DEM), Agency for Data Supply and Efficiency, Copenhagen, Denmark, 2016. https://www.kortforsyningen.dk/sites/default/files/qualityassessmentdk-dem.pdf

Thers, H., Brunbjerg, A. K., Læssøe, T., Ejrnæs, R., Bøcher, P. K., and Svenning, J.-C.: Lidar-derived variables as a proxy for fungal species richness and composition in temperate Northern Europe, Remote Sens Environ, 200, 102–113, https://doi.org/10.1016/j.rse.2017.08.011, 2017.

>> A discussion about the influence of data acquisition season (spring-summer) and the point density (4-5 points/m2) on the final products will be useful for other country's ALS data processing.

Following your comment, we had a closer look at the variation in acquisition time across the point cloud based on the GPS timestamps. To our surprise and contrary to Flatman et al. (2016), the source data contained both leaf off and leaf on data, and also data from 2018, as well as two years before 2014/15 (these are 2007 and 2013). Further enquiries with the Danish Agency for Data Supply and Efficiency (Danish acronym: SDFE) who maintain the source data set were little fruitful. It turned out the publicly available data on kortforsyningen.dk has not been properly versioned by the authorities. Hence, the user (us) ended up with different versions of the data depending on when it is downloaded. Additional ALS surveys were carried out over parts of Denmark in 2018 and 2019 and the publicly available data was updated on the fly without releasing new versions. The dialogue we engaged in with the agency has been very slow and not very helpful. We were pointed to another version of the source data set that is stored on a second data repository maintained by the Danish authorities called daterfordeler.dk. This version of the source data was supposed to only contain data from 2014/15, but again we found that data from 2018 was included albeit for different regions. As we needed to move on with the revision and as it became clear that we were not going to be able to resolve the issue with a simple clean solution, we decided to work with the data we have.

Based on source data sets that are available to us, we re-created the source dataset to contain as much data from 2014/15 as possible, with as little data as possible prior this time period and with no data from later. We ended up using three versions of the source data: One that we downloaded from kortforysningen.dk in April 2020 and used for the initial submission of this data paper, one that we downloaded in September 2021 from datafrodeler.dk and one that was downloaded by a colleague at an unknown time point prior 2017 and used in our previous work published in Moeslund et la. (2019). We documented the data merger in detail on the code repository (source data description: https://github.com/jakobjassmann/ecodes-dk-lidar/blob/master/documentation/source_data/readme.md; description of merger https://github.com/jakobjassmann/ecodes-dk-lidar/blob/master/documentation/source_data/dhm201415_merger.md) and summarised the process in Sect 2.2 (source data), where we also updated Table 1:

**Line 101-109:** Here, we used the DHM/Point-cloud (DHM/Punktsky), the classified georeferenced ALS point cloud product, and the DHM/Terrain (DHM/Terræn), the digital elevation model product derived from the point cloud. The DHM data set is currently maintained by the Agency for Data Supply and Efficiency, Denmark (https://sdfe.dk/) and, at the time of writing, can be downloaded from https://kortforsyningen.dk/ (continuously updated with new survey data) and https://datafrodeler.dk (versioned). While almost all of Denmark's terrestrial surface was covered by ALS surveys in 2014/15, currently none of the products provided by the agency contains data exclusively from these surveys. We therefore merged three different versions of the source data to obtain a dataset that reflects the state of the vegetation in 2014/15 as best as possible, by only containing vegetation data from 2014/15 and limited amounts from 2013 (Table 1, Sect. 3.6.3; see GitHub code repository for a detailed description of the merger and more information on the source data sets).

**Table 1:** Overview of the data sources used for generating the EcoDes-DK15 data set. Three versions of the DHM/Pointcloud were merged to obtain a point cloud data set that contained no vegetation points scanned after 2015 and as little vegetation points before 2014 as possible. DHM/Terrain tiles were matched sources from the same data source as the corresponding point cloud tiles. A copy of the source data is archived on the internal long-term data storage at Aarhus University and is available on request. For further information see documentation on GitHub code repository and Sect. 3.6.3.

| Data source | Years | Used for | Data provider | Downloaded available from (download date) | Number of tiles |
|---|---|---|---|---|---|
| DHM/Pointcloud (DHM/Punktsky) | 2007-2018 | Vegetation Descriptors | Danish Agency for Data Supply and Efficiency | https://kortforsyningen.dk/ (24 April 2020) | 38671 |
| DHM/Pointcloud (DHM2015_punktsky) | 2007-2018 | Vegetation Descriptors | Danish Agency for Data Supply and Efficiency | https://datafordeler.dk (13 September 2020) | 10955 |
| DHM/Pointcloud (GST_2014) | 2007-2015 | Vegetation Descriptors | Danish Agency for Data Supply and Efficiency | https://kortforsyningen.dk/ (unknown, before 2017) | 47 |
| DHM/Terrain (DHM/Terræn) | 2007-2018 | Terrain Descriptors | Danish Agency for Data Supply and Efficiency | https://kortforsyningen.dk/ (24 April 2020) | 38671 |
| DHM/Terrain (DHM2015_terraen) | 2007-2018 | Terrain Descriptors | Danish Agency for Data Supply and Efficiency | https://datafordeler.dk (13 September 2020) | 10955 |
| DHM/Terrain (GST_2014) | 2007-2015 | Terrain Descriptors | Danish Agency for Data Supply and Efficiency | https://kortforsyningen.dk/ (unknown, before 2017) | 47 |

We then reprocessed the tiles from the source datasets we had not already processed using the EcoDes-DK pipeline. We also reprocessed all tiles along the interface between regions from two different sources to ensure no edge effects were created. The outputs were then merged into a new version of the EcoDes-DK15 data set. Scripts and national information for the reprocessing and the merge of the outputs are provided on the code repository (https://github.com/jakobjassmann/ecodes-dk-lidar/blob/master/documentation/source_data/readme.md). The reprocessing required making the EcoDes-DK processing workflow compatible with GDAL 3.3.3 as the GDAL version on our computational server had changed, we therefore made some minor adaptations to the code while keeping backwards compatibility.

**Line 132-133:** Some re-processing was required during the peer review process, for which we used GDAL 3.3.3 from Osgeo4W64 (GDAL/OGR contributors, 2021).

The result is a completely overhauled EcoDes-DK15 data set, version 1.1. Which we have now uploaded to Zenodo (no change to the doi for all versions linked in manuscript, this is still valid: https://doi.org/10.5281/zenodo.4756556).

To inform the reader of the distribution of the survey dates across Denmark and to assist the EcoDes-DK15 end user if they would like to included it as a co-variable in their analysis, we added three new date_stamp* descriptors that summarise the min, max and mode of the survey data for all vegetation points within each 10 m x 10 m cell, based on the GPS time stamps provided in the point cloud. We updated the auxiliary descriptor section of the manuscript accordingly by adding a new section for the date_stamp* descriptors, including a discussion on why the descriptors might be important. We also added an additional figure to help visualise the survey date distribution of the vegetation points in the manuscript (Fig. 5 - all subsequent figures renamed). We further re-generated Fig.4 as the example tile was affected by the reprocessing (new data), adding a note on the effect of flight line overlap on point density that we had previously missed. Lastly, we added to the already existing text in the discussion that deals with the differences in acquisition timing across the pointcloud and updated all summary statistics of the new Eco_Des-DK version across the manuscript.

[revised manuscript text omitted]

**Year, Month**
- 2013, 08
- 2014, 01
- 2014, 03
- 2014, 04
- 2014, 10
- 2014, 11
- 2014, 12
- 2015, 03
- 2015, 04
- 2015, 05
- No date

**Figure 5:** Distribution of the most common survey date for the vegetation points in each tile of the EcoDes-DK15 dataset. The data shown is aggregated for each tile from date_stamp_mode descriptor. The figure highlights that while the majority of the vegetation points is from 2014/15, the data set also includes a small amount of vegetation points from 2013 in western Jutland. Furthermore, surveys were conducted in all seasons, with vegetation points originating in spring, summer, autumn and winter. Nonetheless, the majority of vegetation points comes from the leaf-off season. Lastly, the date_stamp descriptors could not be derived for some regions as the GPSTime was not provided in the point clouds. However, from auxiliary information we know that the surveys in the Mols Bjerge and Sønderborg areas were conducted in April-May 2015 and October 2014, respectively.

(Discussion Section)

**Lines 662-664:** Furthermore, the EcoDes data set is a snapshot in time representing the state of the vegetation in the one and a half years between spring 2014 and summer 2015 (with some exception in western Jutland, where the data is from 2013).

**Lines 667-669:** Additionally, the geographical differences in the timing of the point cloud collection across the country (see Sect 6.3.4) may introduce noise and could affect cross-comparability of the data between regions, for example due to seasonal differences in foliage (see e.g., Leiterer et al., 2015).

(EcoDes-DK Summary Stats)

**Line 20:** (~94 GB, compressed 16.8 GB)

**Line 126-128:** The 1 km x 1 km tiling of the DHM/Terrain 2014/2015 and DHM/Point-cloud data sets 2014/2015 match in extent and geolocation. However, a small number of tiles (n = 30) in the DHM/Point-cloud data sets did not have corresponding tiles in the DHM/Terrain data sets, these were removed prior processing resulting in the total of 49673 tiles shown in Table 1.

**Line 173:** 49673 raster tiles

**Line 178:** The final data set consists of just under 94 GB of data (compressed for download 16.8 GB).

**Line 194:** […] the data set contains nine layers of auxiliary information (see Sect. 3.2).

**Line 203-205:** Processing failed on average for only 18 out of the 49673 tiles per descriptor with a maximum of 65 tiles failing for the canoy_height, normalized_z_mean and normalized_z_sd descriptors.

**Line 609:** (~94 GB)

We also updated the acknowledgements to reflect help we received from our colleagues Zsófia Koma and Matt Barbee:

**Line 743-748:** We would like to thank Andràs Zlinszky for his contributions to earlier versions of the data set, Charles Davison for feedback regarding data use and handling, as well as Matthew Barbee and Zsófia Koma for sharing their insights on the source data merger and Zsófia's script to generate summary statistics for the different versions of the DHM point clouds.

Finally, regarding the suggestion to further discuss the influence of time of acquisition and point density in the manuscript: We can see how a discussion of acquisition timing and point cloud density could be valuable for other countries planning ALS acquisition campaigns. However, we do not think we are in the position to provide a meaningful discussion of these topics in this manuscript, yet make any recommendations that could advise planning of future campaigns based on the data presented in the current manuscript. We were not involved in the planning, logistics and processing of the point cloud, nor do we present any alternative point clouds available to which we could make comparisons (e.g. different acquisition seasons, different point densities densities). That said, we are currently working on a separate project with our colleague Zsófia Koma, who will be addressing these questions based on the different survey campaigns across Denmark (2007, 2014/15, 2018+) which vary in their acquisition timing and point density. So we are optimistic that we will be able to provide an informed discussion on the subject in future publications. In the meantime, other studies targeted at investigating or reviewing these specific questions already cited in the manuscript (e.g., Leiterer et al. 2015 and Bakx et al. 2019) are much better suited in informing such decision making than we are.

Flatman, A., Rosenkranz, B., Evers, K., Bartels, P., Kokkendorff, S., Knudsen, T., and Nielsen, T.: Quality assessment report to the Danish Elevation Model (DK-DEM), Agency for Data Supply and Efficiency, Copenhagen, Denmark, 2016.

Leiterer, R., Furrer, R., Schaepman, M. E., & Morsdorf, F. Forest canopy-structure characterization: A data-driven approach. Forest Ecology and Management, 358, 48–61. https://doi.org/10.1016/j.foreco.2015.09.003, 2015.

Bakx, T. R. M., Koma, Z., Seijmonsbergen, A. C., and Kissling, W. D.: Use and categorization of Light Detection and Ranging vegetation metrics in avian diversity and species distribution research, Divers Distrib, 25, 1045–1059, https://doi.org/10.1111/ddi.12915, 2019.

**Additional corrections (not related to referee comments)**

We removed redundant white spaces throughout the manuscript.

**Line 59:** Corrected typo in 'interest', removing accidental plural 's'.

**Line 64:** Removed 'system' from 'forest system'.

**Line 74:** Changed 'variation of importance to' to 'variation important to'.

**Line 174-175:** We clarified that VRT files are not available for three of the descriptors (point_source_counts, point_source_ids and point_source_proportion), as the creation of VRT files for these multilayer rasters with variable number of layers per tile was not possible: "A [...] (VRT) file is provided for each descriptor (except the point_source_counts, point_source_ids and point_source_proportion descriptors), [...]"

**Line 266:** Dropped article 'The Danish landscapes' now 'Danish landscapes'.

**Line 372, 374, 376 (Figure 3 caption):** Changed 'pixels' to 'cells' for consistency across the manuscript.

**Line 442:** Removed typo: plural 's' from 'cells', now 'cell'.

**Line 535:** Removed typo: repeated full stop, now '.' .

**Line 540:** Corrected grammar: 'in which' now 'that'.

**Line 732:** Removed typo: plural 's' from 'inspirations' not 'inspiration'.

---

## Author Response (AR2)

**Referee #2: Yifang Shi**

Some corrections need to be made before accepting for publication.

Thank you for taking the time to go through our manuscript once more and for providing such thorough feedback. We have made the changes as suggested and documented them line-by-line below.

Line numbers in our replies refer to the revised manuscript.

For example:

Line 15: "remote-sensing" -> "remote sensing"

We have now removed the hyphen in Line 15.

Line 41-43: a "," is missing in the sentence. It should be "If an object intercepting the light pulse is smaller than the beam's footprint (e.g., a leaf or a branch of a tree), some of the light may travel on and trigger a reflection from a second surface (e.g., understory vegetation or the forest floor)."

We have placed the comma as suggested.

Line 253: "32bit" -> "32-bit" or "32 bit", same in line 263 and line 281.

We have added a hyphen to all instances in the manuscript where it was missing between a numerical value and the unit "bit" (Lines: 235, 253, 268, 289, 333 and 579)

Line 239: "the no data value (-9999)", line 276: "the NODATA value -9999", and line 182: "a value of -9999 throughout the data set (NODATA-value)". It is better to describe it in a consistent way.

We have adapted the text. The NoData value is now consistently referred to as "NoData":

Line 182-183: "Missing data (NoData) is denoted by a value of -9999 throughout the data set."
Lines 208-210: "Instead, we generated NoData rasters for all missing descriptor - tile combinations (i.e. we assigned -9999 to all cells in those tiles). We provide a text file listing the affected "NoData" tiles in the folder of each descriptor (the file is named empty_tiles_XXX.txt, where XXX is the descriptor name)."
Line 243: "The cells in those rows and columns have no neighbouring cells and were assigned the NoData value (-9999). "
Line 260-261: "These cells in those rows and columns have no neighbouring cells and gdaldem assigns the NoData value (-9999) to these cells."
Line 281: "The corresponding cells for the affected areas are set to the NoData value -9999."
Line 328-329: "[…] , we set the value of those cells to NoData (-9999). Finally, for cells that are located on the outermost edges of the data set the heat_load_index is not

defined due to propagation of the NoData value assigned to the aspect in those cells."

Line 557: "The masks are binary, a cell value of 1 indicates land and a value of -9999 (NoData) indicates sea or inland water, respectively."

Line 587: "In these cases, we assigned the NoData value to the date_stamps."

Line 313: "an useful" -> "a useful"

Indefinite article changed to "a".

--

In addition to the changes requested by Referee #2 we also changed the font face of "Descriptor" in the first rows of Tables 3 and 4 from normal to bold.

---

## Author Response (AR3)

Thank you for prompt effective changes. We all hope remote sensing and land use communities begin to appreciate and learn utility of airborne lidar data product.

Yes, we hope so too! Thank you for the helpful feedback and coordinating the review process.

Nice use of github (but see small exception below).

Full Zenodo DOI plus 'teaser' link should appear also at end of abstract. Publications staff will insist on these small changes, which can happen during proofing stage.

We have added the links to the Zenodo repository and teaser to the abstract as requested. The final sentence now reads:

The full data set is available on Zenodo: https://doi.org/10.5281/zenodo.4756556 and a 5 MB teaser subset can be found on the GitHub code repository: https://github.com/jakobjassmann/ecodes-dk-lidar/blob/master/manuscript/figure_7/EcoDes-DK15_teaser.zip.

Teaser link (e.g. to .zip file on github) does not work for me; I have a github login but should not need it for this access. Please check and confirm access for users outside of .dk domains.

We apologise for the broken link, the folder structure on the repository changed due to the addition of a figure during the second round of revisions and we forgot to update the teaser link in the main manuscript.

We have updated the link:

https://github.com/jakobjassmann/ecodes-dk-lidar/blob/master/manuscript/figure_7/EcoDes-DK15_teaser.zip

Any user (with or without GitHub account) should now be able to download the file by pressing the download button in the centre right below the GitHub headers. We have confirmed this from IP addresses in Denmark and the UK, but would be grateful if you could verify this also.

We also found one additional link to "https://datafordeler.dk/" in Line 107 (revised manuscript) that was not marked up and have corrected this issue.

Thank you again for using ESSD!

It has been a great experience. Thank you for having us!